# A Catalog of Coding Sequence Variations in Salivary Proteins’ Genes Occurring during Recent Human Evolution

**DOI:** 10.3390/ijms241915010

**Published:** 2023-10-09

**Authors:** Lorena Di Pietro, Mozhgan Boroumand, Wanda Lattanzi, Barbara Manconi, Martina Salvati, Tiziana Cabras, Alessandra Olianas, Laura Flore, Simone Serrao, Carla M. Calò, Paolo Francalacci, Ornella Parolini, Massimo Castagnola

**Affiliations:** 1Dipartimento Scienze della Vita e Sanità Pubblica, Università Cattolica del Sacro Cuore, 00168 Rome, Italy; lorena.dipietro@unicatt.it (L.D.P.);; 2Fondazione Policlinico Universitario A. Gemelli IRCCS, 00168 Rome, Italy; 3Laboratorio di Proteomica, Centro Europeo di Ricerca sul Cervello, IRCCS Fondazione Santa Lucia, 00179 Rome, Italy; 4Dipartimento di Scienze della Vita e Dell’ambiente, Università di Cagliari, 09042 Monserrato, Italy; 5Department of Medicine and Surgery, Proteomics and Metabolomics Unit, University of Milano-Bicocca, 20854 Vedano al Lambro, Italy

**Keywords:** salivary proteins, nucleotide substitutions, evolution

## Abstract

Saliva houses over 2000 proteins and peptides with poorly clarified functions, including proline-rich proteins, statherin, P-B peptides, histatins, cystatins, and amylases. Their genes are poorly conserved across related species, reflecting an evolutionary adaptation. We searched the nucleotide substitutions fixed in these salivary proteins’ gene loci in modern humans compared with ancient hominins. We mapped 3472 sequence variants/nucleotide substitutions in coding, noncoding, and 5′-3′ untranslated regions. Despite most of the detected variations being within noncoding regions, the frequency of coding variations was far higher than the general rate found throughout the genome. Among the various missense substitutions, specific substitutions detected in *PRB1* and *PRB2* genes were responsible for the introduction/abrogation of consensus sequences recognized by convertase enzymes that cleave the protein precursors. Overall, these changes that occurred during the recent human evolution might have generated novel functional features and/or different expression ratios among the various components of the salivary proteome. This may have influenced the homeostasis of the oral cavity environment, possibly conditioning the eating habits of modern humans. However, fixed nucleotide changes in modern humans represented only 7.3% of all the substitutions reported in this study, and no signs of evolutionary pressure or adaptative introgression from archaic hominins were found on the tested genes.

## 1. Introduction

Saliva is a multifaceted bodily fluid that contains enzymes (amylases, lysozymes, and lipases), proteins, peptides and glycoproteins, lipids (hormones such as testosterone and progesterone), and proteases, along with a high concentration of inorganic ions [1]. To date, more than 2000 proteins and peptides have been identified in saliva [2]. They are mainly involved in the homeostasis of the oral cavity, the digestion process, and the innate immune response [3]. Ninety percent of the salivary proteins and peptides derive from the secretion of the three major salivary glands (parotid, submandibular, and sublingual glands), while the remaining 10% are secreted by minor salivary glands or derive from exfoliated cells and leucocytes present in the gingival–crevicular fluid [4] from plasma exudate, plus some contributions from the oral microbial flora. During their transit in the secretory pathway, salivary proteins undergo a series of post-translational modifications (PTMs), including phosphorylation, N-terminal acetylation, glycosylation, sulfation, and proteolytic cleavages. Further changes in proteins and peptides also occur after secretion in the oral cavity, through the action of exogenous (microflora) and endogenous enzymes [1].

The main contribution to the composition of the human salivary proteome derives from a few protein families. In particular, proline-rich proteins (PRPs), statherin (STATH), P-B peptide, histatins (HTN), cystatins (CST), and amylases (AMY) altogether represent more than 95% (*w*/*w*) of all proteins found in saliva to date [5]. PRPs represent the major fraction of the salivary proteome in *Homo sapiens* (nearly 70% of the total protein content; >50% in weight) and include basic (bPRPs), acidic (aPRPs), and basic glycosylated (gPRPs) PRPs. They share a high abundance of proline, glycine, and glutamine residues, which represent 70–80% of the entire amino acid sequence [6,7]. bPRPs include eleven parent peptides/proteins and more than six parent glycosylated proteins (gPRPs), plus several proteoforms derived from gene polymorphisms and PTMs [8,9,10] (Figure 1). PRPs are encoded by genes belonging to the PRP multigene family, located within the *PRB* locus mapping on 12p13.2. The locus includes six tandemly linked genes: *PRB2*–*PRB1–PRB4–PRH2–PRB3*–*PRH1*, in the 5′-to-3′ direction, and is highly polymorphic as it contains internally repetitive DNA sequences, leading to frequent recombinational events [11,12]. At least four alleles (S, small; M, medium; L, large; and VL, very large) are present in the Western population of *Homo sapiens* at *PRB1* and *PRB3* loci and three (S, M, L) at *PRB2* and *PRB4* loci [8] (Figure 1). Except for the protein encoded by the *PRB3* locus that gives rise to gPRPs, all the bPRP pro-proteins are cleaved completely by pro-protein convertases, generating smaller peptides/proteins, before granule maturation [9] (Figure 1). aPRPs are expressed in two loci, *PRH1* and *PRH2*, mapping on chromosome 12p13. Single amino acid substitution and repeat insertion generate three *PRH1* alleles, encoding parotid isoelectric-focusing slow isoform (PIF-s), the parotid acidic protein (Pa)—both 150 residues long—and the double band isoform slow (Db-s)—171 amino acid residues long [10] (Figure 2A). A single nucleotide substitution generates two *PHR2* alleles, encoding the PRP-1 and PRP2 isoforms [11] (Figure 2A). A pro-protein convertase partially cleaves PRP-1, PRP2 and PIF-s in 3 N-terminal fragments of 106 residues, called PRP3, PRP4, PIF-f (PRP3 type), and a common C-terminal fragment of 44 amino acids, called P-C peptide. Db-s is cleaved at position 127 generating two peptides: Db-f (f stands for fast) and the P-C peptide (same as above) [12] (Figure 2A). The Pa isoform not carrying the convertase sequence generates a dimeric form through a disulfide bond [13] (Figure 2A). STATH is encoded by the *STATH* gene located in chromosome 4q13-19 [13,14]. Several STATH proteoforms are detectable in saliva due to phosphorylation, cyclization by transglutaminase 2, and proteolysis by amino-/carboxy-peptidases and convertase action [13,15,16]. P-B is a proline-rich small peptide encoded by the *SMR3B* gene, mapping on chromosome 4q13.3 [17], near the *STATH* gene, possibly sharing epigenetic control and/or the DNA replication timeframe [13,15,16]. HTN are small cationic histidine-rich peptides encoded by the *HTN1* and *HTN3* genes on chromosome 4q13. Despite their high sequence homology, HTN1 and HTN3 have different maturation pathways and biological activities [17,18,19].

CST are inhibitory cysteine proteases involved in the innate immune response [20]. CSTA and CSTB are encoded by *CSTA* and *CSTB* genes, respectively, whereas CST-SN, CST-SA, CST-C, CST-S, and CST-D are encoded by *CST1*-*CST5* genes (Figure 2B). Several PTMs occur in CST proteins, including N-acetylation, proteolytic cleavages, phosphorylation, and M-, W-, and C-oxidation, causing different final protein structures detectable in human saliva [21]. Also, two isoforms generated by single amino acid substitutions of cystatin D and cystatin SN are present in saliva [21] (Figure 2B).

The amylase alpha 1A (*AMY1A*) gene, on chromosome 1p21.1, is responsible for the expression of AMY, which accounts for about 20% of the weight of salivary proteins and is the most abundant protein of the whole saliva of *Homo sapiens*.

Several comparative studies have shown that the human salivary proteome differs from other species due to genetic divergences that are possible due to environmental factors, including diet and pathogens [22,23,24,25]. A recent study reported the results obtained from the comparison of the salivary proteomes of *Homo sapiens sapiens* (modern humans) with our closest extant evolutionary relatives, chimpanzees, and gorillas [26]. The authors demonstrated that the salivary protein composition is unique to each species despite their close sequence homology, which likely reflects an evolutionary adaptation [26]. Despite this initial observation, the evolution of human loci-encoding salivary proteins has not been studied to date. Nowadays, the increasing amount of genomic data obtained through sequencing of preserved skeletal remains of extinct hominins, such as *Homo neanderthalensis* (Neanderthals) and *Homo Denisova* (Denisovans), can reveal the extent of diversity that has emerged at the genomic level during more recent human evolution.

In this study, we aimed to identify the sequence changes that have been fixed during the recent human evolution in the gene loci encoded for the most abundant salivary proteins (namely, PRPs, statherin, P-B peptide, histatins, cystatins, and amylases) to gather possible functional indications regarding their evolutionary path and their contribution to oral homeostasis and salivary functions. Eating habits may be indeed mutually implicated with salivary proteins’ biology since these are implicated in the modulation of the microbiome of the oral cavity and the entire gastrointestinal tract [26]. To achieve this, we have interrogated the publicly available sequence databases of Neanderthals and Denisovans and compared them with modern human genome sequence data. This allowed us to identify several nucleotide substitutions in the loci coding for the most relevant human salivary protein families.

## 2. Results

By comparing the genomic sequences of salivary gene loci in modern humans with those of Altai Neanderthals, Chagyrskaya Neanderthals, Vindija Neanderthasl, and Denisovans, we identified an overall number of 3472 sequence variants/nucleotide substitutions across the 17 tested salivary genes in coding, noncoding, 5′-3′ untranslated (UTRs), and regulatory regions. The nucleotide substitutions observed in the 17 salivary-tested genes were summarized in Figure 3. Of the 3472 changed nucleotides, only 428 were in coding regions, and 121 were annotated as synonymous (Figure 3). The remaining 307 nucleotide variations were nonsynonymous (Figure 3), which are known to be subjected to a higher evolutionary pressure and are frequently exposed to natural selection [27,28]. We have, therefore, attempted a functional interpretation of nonsynonymous variations, which is inherently speculative and deserves future functional studies. The potential impact of nonsynonymous variants on salivary proteins’ function of Neanderthals and Denisovans was predicted by a SIFT (sorting intolerant from tolerant) analysis (see Table 1, Table 2 and Table 3), which enables predicting amino acid substitutions that may exert a deleterious effect. The reference single nucleotide polymorphism (SNP) number (rs) and the corresponding frequencies of the 107 missense changes in coding regions were also reported in Table 1, Table 2 and Table 3. Of note, even though the nucleotide changes located in noncoding regions should not affect the primary structure of the encoded protein, they could affect regulatory elements that may modify the splicing and/or the binding of epigenetic modulators and/or chromatin folding/looping. The variants fixed at 100% in modern humans compared to ancient hominines were highlighted in light orange in Table 1, Table 2 and Table 3 and Appendix A.

In the following subparagraphs, the results were detailed considering one locus at a time. Note that given the extreme structure heterogeneity of the tested genes with multiple alleles and different lengths, the nucleotide variations were indicated according to their genomic coordinates (see Section 4 for details).

### 2.1. Nucleotide Variations in the Gene Loci Encoding Basic Proline-Rich Proteins

#### 2.1.1. *PRB1* Gene

The genomic alignment allowed us to identify 130 nucleotide changes in the *PRB1* gene in ancient hominines compared with modern humans (Table 1 and Appendix A). Fifty-five of these were detected within coding exons and included ten synonymous and forty-five nonsynonymous nucleotide substitutions. Among the nonsynonymous nucleotide substitutions, 20 corresponded to SNPs annotated in modern humans (Table 1). SIFT prediction indicated that 46% of these missense variants have a significant effect on protein function based on sequence homology and the physical properties of the involved amino acids (Table 1). The T-C transition, which occurred in modern humans at position 11,506,774, causing the substitution of R_72_ with a Q in the II-2 isoform (Table 1 and Figure 4a), may have an impact on post-translational protein processing. Indeed, the modern human R_72_ residue is part of the R_72_SPR_75_ consensus sequence recognized by the pro-protein convertase responsible for the cleavage between II-2 and P-E peptides. Therefore, we may hypothesize that in archaic species, the PRB-1-encoded protein was a fused peptide spanning 136 amino acids, which integrates the modern II-2 and P-E (Table 1 and Figure 4a). The sequences of the peptides and the resulting putative archaic protein primary structures (named PRB-1 salivary archaic fusion 1 peptide, PRB-1 SAF-1) are reported in Figure 4a. The remaining seventy-five nucleotide changes identified in the *PRB1* locus were found to fall within noncoding regions, namely fifty-four in introns, six in upstream regions, one in the 5′ UTR, 1 in the 3′UTR, and thirteen in downstream regions (Appendix A).

#### 2.1.2. *PRB2* Gene

One hundred and thirty-six nucleotide substitutions were detected in the *PRB2* locus in ancient hominines compared with modern humans (Table 1 and Appendix A). Thirty-seven of these were identified in introns, ten in upstream regions, one in the 3′UTR, and eight in downstream regions. The remaining eighty variations were found in coding regions, namely two in exon 1 (corresponding to the signal peptide), one in exon 2, and the remaining in exon 3 (Table 1 and Appendix A). Of note, the modern human sequence reported in the UniProtKB database corresponded to the L allele coding for the common isoforms IB-8a Con1^-^ and P-H S_1_, the first one with a P residue instead of an S at position 100, the second one with an S residue instead of an A at position 1 [8]. Of the 80 sequence variants found in coding exons, 64 were nonsynonymous, causing amino acid substitutions. SIFT prediction indicated that 19% of these missense variants have a significant effect on protein function based on sequence homology and the physical properties of the involved amino acids (Table 1). Twenty-six out of the sixty-four nonsynonymous substitutions were annotated as common variants (SNPs) in modern humans (Table 1). In particular, two changes occurring at 11,546,686 bp and 11,546,677 bp caused the substitution of the R_93_ and R_96_ with Q within the ancient IB-1 isoform. The two archaic residues were found in all four species, (Table 1). This implied that the archaic hominins’ R_93_SPR_96_ consensus sequence, recognized by the pro-protein convertase, apparently lacked two key arginine residues, thus disabling the post-translational cleavage. Therefore, the ancient saliva composition should feature a protein deriving from the fusion of IB-1 and P-J peptides, spanning 157 amino acids (named the PRB-2 salivary archaic fusion 2 peptide, PRB-2 SAF-2 peptide, in Figure 4b). Conversely, the presence of a C nucleotide at 11,546,314 bp in Neanderthals and Denisovans, instead of T in modern humans, led to the introduction of an R instead of the Q_59_ (Q_217_ in pro-protein) of the IB-8a Con1^-^ isoform. This archaic primary structure would then include an additional pro-protein convertase consensus sequence, R_59_SAR_62_, causing the cleavage of the IB-8a Con1^-^ protein into two smaller peptides. According to the usual removal of the C-terminal arginine residue observed for almost all the bPRPs, both peptides should be 61 aminoacidic residues long (Figure 4c). These putative archaic hominins’ PRB-2 variants are named by us the PRB-2 salivary archaic cleavage 1 peptide (PRB-2 SAC-1 peptide) and the PRB-2 salivary archaic cleavage 2 peptide (PRB-2 SAC-2 peptide) and are shown in Figure 4c. Of note, the sequence of the PRB-2 SAC-1 peptide exactly corresponds to the sequence of the modern human P-J peptide with an alanine (A_61_) instead of a serine in the last amino acid residue. The sequence of the PRB-2 SAC-2 peptide exactly corresponds to the modern human P-F peptide with a serine (S_61_) instead of an alanine in the last amino acid residue (Figure 4d and [9]). The variation at 11,546,395 bp indicated that in archaic hominins, the P_31_ (P_189_ of pro-protein) residue was replaced by a Q in the IB-8a Con1^-^; this change results probably in a deleterious effect on protein function, as predicted by SIFT analysis.

The protein name, the modifications with respect to modern humans, and the corresponding frequencies found in Neanderthals, Chagyrskayas, Vindijas and/or Denisovans are reported for each archaic protein. The positions of each substitution are also reported in the primary sequences (residues in bold characters). q: pyroglutamic acid; S: phosphorylated serine.

#### 2.1.3. *PRB3* Gene

We have identified 163 nucleotide variations in the *PRB3* locus in ancient hominines compared with modern humans (Table 1 and Appendix A). Of these, 53 were detected in coding regions and 110 in noncoding regions (71 within introns, 14 in upstream regions, 2 in the 3′UTR, and 23 in downstream regions; Appendix A). The archaic sequences were compared with the allele Gl-2 (or PRP-3M) of modern humans. Fourteen variations identified in coding exons were synonymous, whereas thirty-nine changes were missense variants. Twelve out of the thirty-nine nonsynonymous substitutions corresponded to annotated common variants in modern humans (Table 1). PRP3 protein contains eight N-glycosylated Asp residues falling into the NXS/pS sequon; among the substitutions found in the *PRB3* gene, only those at position 11,420,728 fall within the consensus sequence (S_136_F), and deleterious results for the protein function were predicted by SIFT (Table 1). Overall, 37.5% of the substitutions were found to be deleterious on the protein function (Table 1). The noncoding variant found at position 11,420,458 could probably affect the splicing process of *PRB3* transcripts in ancient hominins since it fell within the GU consensus site (splice donor site) at 5′ end of intron 3 (Appendix A).

#### 2.1.4. *PRB4* Gene

For the *PRB4* locus, we detected 129 nucleotide substitutions in ancient hominines compared with modern humans (Table 1 and Appendix A). Of these, 27 were found in coding exons, including 4 synonymous and 23 nonsynonymous (Table 1), and 102 in noncoding regions (Appendix A). The archaic sequence was compared with the small allele of the modern human locus coding for P-D peptides and glycosylated protein A (PGA). The 23 missense variants were all found within coding regions for the glycosylated protein A, while none of the identified variations would affect the P-D variant (see Table 1 for details). These variations had no consequence on the consensus sequence of pro-protein convertase or on the sequence of the glycosylation sites. It is interesting to observe that all the archaic sequences reported a code for the P-D P_32_A variant. Overall, seven out of the twenty-three nonsynonymous in the *PRB4* locus corresponded to annotated common variants in modern humans, and only 13% were found to be deleterious on the protein function (Table 1).

### 2.2. Nucleotide Variations in the Gene Locus Encoding the a-PRP

One hundred and sixty-three nucleotide substitutions have been annotated in the *PRH2* gene locus in ancient hominines compared with modern humans (Table 2 and Appendix A), of which thirty fell within coding exons, including seven synonymous and twenty-three nonsynonymous. Four of these latter corresponded to annotated common variants in modern humans (Table 2). Sixty-six nucleotide substitutions were identified in introns, seven in upstream regions, three in the 5′UTR, forty-nine in the 3′UTR, and eight in downstream regions (Appendix A). The archaic DNA sequences reported in the sequence database used in this study (see Section 4 for details) corresponded to the PRP-1 protein of the *PRH2* alleles, thus having a N_50_ residue. The nucleotide variations reported in Table 1 generated two synonymous substitutions at D_6_ and P_135_.

### 2.3. Nucleotide Variations in the HTN Gene Loci

A total of 188 and 175 nucleotide substitutions were identified in the *HTN1* and *HTN3* genes, respectively (Table 2, Appendix A). The nucleotide substitutions reported in *HTN1* are distributed as follows: 4 fell within coding exons, including1 synonymous and 3 nonsynonymous, and 184 fell in noncoding regions, including146 within introns, 6 in upstream regions, 3 in the 5′UTR, 9 in the 3′UTR, and 20 in downstream regions (Table 2 and Appendix A). Regarding *HTN3*, 3 nucleotide changes were reported in coding exons (1 synonymous and 2 nonsynonymous), whereas 172 fell in noncoding regions (145 within introns, 9 in upstream regions, 3 in the 5′UTR, 5 in the 3′UTR, and 10 in downstream regions) (Table 2 and Appendix A). One missense variant for *HTN1* and one for *HTN3* found in ancient hominins were also reported as SNPs in modern humans (Table 2).

### 2.4. Nucleotide Variations in the AMY1A Gene Locus

Two hundred and twelve nucleotide substitutions have been annotated in the *AMY1A* gene locus in Neanderthals and Denisovans compared with modern humans (Table 2 and Appendix A). Forty changes fell within coding exons, of which eleven were synonymous and twenty-nine were nonsynonymous. Only one of the nonsynonymous substitutions corresponded to an annotated common variant in modern humans (Table 2). One hundred forty-four nucleotide substitutions were identified in introns, four in upstream regions, nine in the 5′UTR, and fifteen in downstream regions (Appendix A).

### 2.5. Nucleotide Variations in the STATH and P-B Gene Loci

One hundred fifty-nine nucleotide substitutions have been annotated in the *STATH* gene locus in Neanderthals and Denisovans compared with modern humans (Table 2 and Appendix A). Six changes fell within coding exons, of which two were synonymous and four were nonsynonymous (Table 2). One hundred fifty-three nucleotide substitutions were detected in introns and regulatory regions (Appendix A).

One hundred eighty-seven nucleotide substitutions were detected in the *SMR3B* locus in Neanderthals and Denisovans compared with modern humans (Table 2 and Appendix A). Of these, 5 were found in coding exons (2 synonymous and 3 nonsynonymous), 155 were in introns, 3 in upstream regions, 3 in 5′UTRs, 10 in 3′UTR, and 11 in downstream regions (Table 2 and Appendix A). One missense variant was reported as an SNP in modern humans (Appendix A).

### 2.6. Nucleotide Variations in the CST Gene Loci

#### 2.6.1. *CST1* Gene

We have annotated 227 nucleotide substitutions in the *CST1* locus in Neanderthals and Denisovans compared with modern humans (Table 3 and Appendix A). Of these, 128 were found in introns, 19 in upstream regions, 7 in the 5′UTR, 12 in the 3′UTR, 32 in downstream regions (Appendix A), and 29 in coding regions, including 11 synonymous and 18 missense variations (Table 3). The nucleotide variation at 23,731,494 bp caused the substitution of the Y_3_(sp) with an H, affecting the third amino acid residue of the signal peptide. This should not impact the function of the protein, although it may have affected the speed of protein translation and/or the correct processing and trafficking. Four substitutions out of eighteen could have a negative impact on protein function, as predicted by SIFT. Overall, nine nonsynonymous nucleotide substitutions corresponded to annotated common variants in modern humans (Appendix A).

#### 2.6.2. *CST2* Gene

We detected 167 nucleotide changes in the *CST2* locus in Neanderthals and Denisovans compared with modern humans (Table 3 and Appendix A). Of these, 103 were in introns, 15 in upstream regions, 8 in the 3′UTR, 17 in downstream noncoding regions (Appendix A), and 24 in coding regions (Table 2). The latter included six synonymous and nineteen nonsynonymous variations, eight of which were predicted to have a deleterious effect on protein function (SIFT score < 0.05). Ten out of the eighteen nonsynonymous substitutions corresponded to annotated common variants in modern humans (Table 2). Interestingly, the nucleotide change at 23,804,691 bp fell into the canonical DNA-binding motif for the NR3C1 (nuclear receptor subfamily 3 group C member 1) transcription factor, as reported in the UCSC Genome Browser. This variation could most likely affect the affinity of this factor for the regulatory region and thus the expression of the *CST2* gene.

#### 2.6.3. *CST3* Gene

In the *CST3* locus, we have identified 452 nucleotide variations in Neanderthals and Denisovans compared with modern humans (Table 3 and Appendix A). Of these, 329 were in introns, 18 in upstream regions, 9 in 5′UTR, 50 in 3′UTR, 29 in downstream noncoding regions (Appendix A), and 17 in coding regions, including 9 synonymous and 8 nonsynonymous variations (Table 2). One nucleotide substitution corresponded to an annotated common variant in modern humans (Table 2).

#### 2.6.4. *CST4* Gene

Two hundred and sixty-three nucleotide substitutions were detected in the *CST4* locus in Neanderthals and Denisovans compared with modern humans (Table 3 and Appendix A). These included 130 changes in introns, 42 in upstream regions, 4 in the 5′UTR, 20 in the 3′UTR, 43 in downstream noncoding regions (Appendix A), and 24 in coding exons (11 synonymous and 13 missense variations; Table 3). Seven variations in this locus corresponded to annotated common variants in modern humans (Table 3). The change at 23,666,565 bp caused the substitution of the M_111_ with an R in the corresponding Neanderthal peptide structure. Even if it causes the substitution of an uncharged amino acid with a charged one, the SIFT analysis did not predict a deleterious effect of this variant on the function of the archaic protein compared to modern humans.

#### 2.6.5. *CST5* Gene

One hundred ninety-three nucleotide substitutions were annotated in the *CST5* locus in Neanderthals and Denisovans compared with modern humans (Table 3 and Appendix A). Sixteen changes were mapped in the coding region, including eight synonymous and eight nonsynonymous (Table 3). Of the 177 nucleotide substitutions located in noncoding regions, 118 were in introns, 24 in upstream regions, 18 in 3′UTR, and 17 in downstream regions (Appendix A). The exonic nucleotide variation generated the codon for an R in both archaic hominins instead of C_26_. This represented a common variant also found in modern humans (rs1799841). The cystatin D variant with the R_26_ is frequently detected in the soluble fraction of human saliva, probably because is more soluble than the C_26_-containing isoform [19]. Moreover, the opposite substitution (R_26_C) was detectable with high frequency at the same amino acid residue in the cystatin SA gene of Neanderthals. Five out of the eight nonsynonymous nucleotide substitutions corresponded to annotated common variants in modern humans (Table 3).

#### 2.6.6. *CSTA* and *CSTB* Genes

Finally, 394 and 134 nucleotide substitutions were identified in *CSTA* and *CSTB* loci, respectively, in Neanderthals and Denisovans compared with modern humans (Table 3, Appendix A). The nucleotide substitutions reported in *CSTA* were distributed as follows: 6 fell in coding exons, including 2 synonymous and 4 nonsynonymous, and 388 fell in noncoding regions, including 346 in introns, 10 in upstream regions, 5 in the 5′UTR, 10 in the 3′UTR, and 17 in downstream regions (Table 3 and Appendix A). Among these changes, the variation at 122,044,848-122,044,850 positions of *CSTA* was a CTT deletion, observed exclusively in Denisovans (Appendix A). This fell within the canonical DNA-binding motif for the Spi-1 proto-oncogene transcription factor (source: UCSC Genome Browser); therefore, it could probably affect the expression of the *CSTA* gene in the ancient hominin. Regarding *CSTB*, 9 nucleotide changes were reported in coding exons (6 synonymous and 3 nonsynonymous), whereas 125 fell in noncoding regions (55 within introns, 27 in upstream regions, 5 in the 5′UTR, 15 in the 3′UTR, and 23 in downstream regions) (Table 3 and Appendix A). One missense variant for *CSTA* and 1 for *CSTB* found in ancient hominins were also reported as an SNP in modern humans (Table 3).

### 2.7. Geographic Distribution of Genetic Variants in Modern Humans

Of note, the salivary protein genes tested resulted polymorphic in humans. The frequency of specific coding nonsynonymous genetic variants also changed between different populations, as reported in the Geography of Genetic Variants Browser (https://popgen.uchicago.edu/ggv; accessed on 22 July 2022) (Appendix A) [29]. In particular, 20 genetic variants (three in the *PRB1* gene, six in *PRB2*, one in *PRB3*, two in *CST1*, four in *CST2*, three in *CST5*, and one in *CSTB*; highlighted in red in Table 1, Table 2 and Table 3) displayed a different geographic distribution and specifically; rs554211998, rs201994479, rs34305575, rs6076122, rs111349461, rs55860552, rs568411970, rs145031249, and rs1799841 showed a peculiar allele frequency in African populations (Appendix A).

### 2.8. Evolutionary Pressure of Salivary Protein Genes

To investigate if some of the salivary protein genes studied showed evidence of positive selection in anatomically modern humans, we performed a population branch statistics (PBS) analysis [30]. Our results showed no signal of recent selective pressure for the genes analysed, attesting that variants on these genes did not affect individual fitness (Appendix A). We also implemented the Tajima test as an additional evolutionary analysis to evaluate the selective effects of each observed substation. Tajima’s D values show comparable variance among the genes analysed. The D values were prevalently slightly negative or positive (ranging from −0.698 to 3.359) (Appendix A), confirming the absence of a selective sweep [31], which was already suggested by the PBS test.

Compared to modern humans, Neanderthal and Denisovan genomes showed evidence of ancient interbreed [32], leading to an uneven distribution of introgressed chromosomal regions because of natural selection [33]. To investigate if some of the salivary protein gene variants studied might be due to interbreeding, we used two databases of archaic introgression based on a comparison with modern genomes from the 1000 genomes project [34] and the Estonian Biocentre collection [35], which also reported data from previous studies [33,36]. However, the considered genes were not encompassed within the chromosomal regions highlighted in the databases and, therefore, did not show an apparent sign of adaptative introgression from archaic hominins.

## 3. Discussion

The different dietary habits of archaic hominins and modern humans have been mostly attributed to the changes in the availability of natural food resources, the oral bacterial community (microbiota), and climatic conditions [37,38]. A role for salivary proteins can be also inferred, as they are known to be implicated in the modulation of the microbiome of the oral cavity, the entire gastrointestinal tract, and taste perception [39]. aPRPs can promote the attachment of several important bacteria, such as *Actinomyces viscosus*, *Bacteroides gingival*, and some strains of *Streptococcus mutans.* Moreover, both aPRPs and statherin promote the colonization of oral surfaces by *Porfiromonas gingivalis* [40]. It was reported that the salivary proteins may modulate oral health and homeostasis, maintain a stable ecosystem, and inhibit the growth of cariogenic bacteria [41,42]. Recently, 258 salivary proteins were found differentially expressed between the caries-free and caries-active children [43]. They are also involved in taste perception. In particular, the salivary bPRPs II-2 and Ps-1 contribute to bitter taste sensitivity [44]. Also, some salivary peptides belonging to the bPRPs and the histatin families can bind polyphenols in tannin-rich foods, thus evoking the typical astringent sensation [44]. Salivary proteins play an important role in affecting sweet [45], salt [46], and umami [47] tastes, along with fat, salt, and bitter acceptance [48,49]. Also, cystatins are supposed to affect taste perception, as lower salivary levels of these peptides may enhance proteolysis, which would affect the mucosal pellicle lining of the oral cavity, thereby increasing the accessibility of tastants to taste receptors [49]. Interestingly, most of these proteins have been shown to be modulated in pathological conditions, including tumors and inflammation, suggesting that they play a role as clinically relevant biomarkers [5].

Therefore, a hypothesis has been raising that the evolutionary changes occurred in the structure of these proteins could be associated with the different dietary habits of archaic hominins. In this regard, mutations in different bitter taste receptor genes (namely *TAS2R62*, *TAS2R64*, and *TAS2R38*) and the masticatory myosin gene *MYH16*, along with the duplication of the salivary amylase gene *AMY1* that has occurred in recent human evolution, have been associated with variations in taste sensitivity and the shift toward the food cooking habits of modern humans [50].

Based on this emerging background, in this study, we identified and inferred the functional consequences of the nucleotide substitutions fixed in the gene loci coding for the main salivary proteins in modern humans compared to ancient hominins species (Neanderthals and Denisovans).

By mapping over 3400 nucleotide substitutions, we have shown that the majority (87.7%) of changes are detectable in the genes expressing the most important salivary proteins (proline-rich proteins, statherin, P-B peptides, histatins, cystatins, and amylases) of modern humans, compared with Neanderthals and Denisovans, mapped within noncoding regions.

Quite unexpectedly, our data also showed the presence of nucleotide variations affecting the coding sequence of all 17 gene loci analysed. Overall, the frequency of coding variations in these genomic loci is far higher than the general rate found throughout the genome since previous studies highlighted that relatively few amino acid changes have become fixed in recent human evolution to date [51,52]. To the best of our knowledge, this study provides the first original description of coding nucleotide changes that occurred in salivary protein genes during the recent evolutionary shift of modern humans from Neanderthal and Denisovan species. Focusing on these missense variations, we hypothesized the possible functional effects they could have played in protein structure, processing, and function. Of the 307 missense changes found in the coding regions of the tested genes, 92 were predicted to have a potentially deleterious effect on protein function.

The changes identified in the *PRB1* and *PRB2* genes are worth particular attention and could be interpreted in light of the extant knowledge of the biology of the encoded proteins. As already mentioned, the PRB protein family is highly polymorphic and, despite being common to all mammals, the proteins belonging to this family feature have significant structural differences among species. For instance, the peptides generated by the convertase cleavage span 50 to 90 amino acids in length in humans and 10 to 40 in pigs, with sensible variations in the peptide sequences [53]. Therefore, bPRPs appear to be non-conserved across species, probably because they are mostly implicated in taste perception and underwent a deep transformation during evolution due to the changing habits and habitats of the species [44]. Interestingly, our results showed that three nucleotide substitutions annotated in the archaic hominins’ *PRB1* and *PRB2* genes affect specific arginine residues within the consensus sequences of the polypeptide, which are recognized by the pro-protein convertases responsible for their cleavage. These changes could have determined the presence of fused proteins in the archaic hominins’ proteome. The putative “PRB1 salivary archaic fusion 1 peptide” and “PRB2 salivary archaic fusion 2 peptide” could have been possibly associated with additional and/or alternative functions that able to influence the eating habits of extinct hominins. In addition, we have also identified a sequence change in the *PRB2* gene that instead generates a new pro-protein convertase consensus sequence in the encoded peptide. As a result, ancient hominins could have expressed two smaller peptides, the “PRB2 salivary archaic cleavage 1 peptide” and the “PRB2 salivary archaic cleavage 2 peptide”, possibly exerting alternative functions, which deserve further functional studies.

The missense nucleotide substitutions annotated in the remaining salivary protein genes described in this study (aPRPs, histatins, amylases, statherin, P-B peptide, and cystatins) could be interpreted, at least in part, considering the putative changes that they can cause in post-translational protein processing, sorting, localization, and trafficking toward secretion. In addition, all the missense variations that introduce or remove a cysteine residue on the archaic cystatins, most likely affecting the conserved sequences involved in the protein-protein binding [53], could also influence protein function.

We also annotated the nucleotide variations fixed within the noncoding regions of modern humans of the tested genes, given these could reasonably affect the expression levels of salivary proteins by changing the affinity of transcriptional regulators for promoters, enhancer and/or silencer elements, and/or the splicing, in addition to changing splice site consensus sequences and leading to the formation of alternative coding transcripts. Also, they could affect post-transcriptional regulation mechanisms, such as the binding of the noncoding regulatory RNAs, leading to varying protein types and amounts that emerged during the recent evolution. Specifically, two nucleotide substitutions found in the *CST2* and *CSTA* gene loci appear to fall within the canonical DNA-binding motifs for specific transcriptional factors, which could most likely intervene in the modulation of their expression. We also annotated 216 changes in the 3′ untranslated regions in 16 of the 17 genes analysed (in all but *AMY1A*). These substitutions might instead condition the binding of specific microRNA-targeting salivary protein transcripts, modulating their stability and the translation process.

Lastly, 34.9% of the nonsynonymous nucleotide substitutions identified in this study appear to be frequent in the modern human genome, where they are annotated as single nucleotide polymorphisms (SNPs). In addition, some of these coding genetic variants display a different geographic distribution in humans. This observation reduces the evolutionary significance of such changes, which are to be considered in light of the polymorphic nature of these genomic loci. However, taken together, variants showing alternative nucleotide fixation in modern vs. archaic humans represent 7.3% of all the nucleotide substitutions reported in the study.

Also, our results do not suggest any significant evolutionary pressure or sign of adaptative introgression from archaic hominins on the tested genes.

## 4. Materials and Methods

### 4.1. Nucleotide Variants Annotation

In order to annotate all the nucleotide variants within the gene loci of the salivary proteins of interest, we compared modern human sequences with Altai Neanderthals (downloaded from http://cdna.eva.mpg.de/Neanderthal/altai/AltaiNeanderthal/bam/, accessed on 2 May 2020), Chagyrskaya Neanderthals (Index of/neandertal/Chagyrskaya/BAM (mpg.de), accessed on 9 December 2022), Vindija Neanderthals (Index of/neandertal/Vindija/bam/Pruefer_etal_2017/Vindija33.19 (mpg.de), accessed on 9 December 2022), and Denisova sequences (http://cdna.eva.mpg.de/denisova/alignments/, accessed on 2 May 2020) [54,55]. The fossil remains, aged between 50,000 and 30,000 years, come from two distinct geographical areas. The female Neanderthal sample from Vindija (Croatia), in the Western Balkans, yielded a 30× genome coverage [56]. The other samples came from two different sites in the Altai Mountains in Siberia (Russia): the genomic data of a female Neanderthal (at 52× coverage) [57] and a juvenile female Denisovan individual (at 30× coverage) [55] came from the Denisova cave, and another female sample came from the Chagyrskaya cave, located about 100 km westward, and yielded a genome of 27× coverage [58]. In particular, we aligned the sequences of modern humans and ancient hominines by means of the Integrative Genomics Viewer (IGV) tool (2.3.72 version) [59,60,61]. Note that the reference genomes annotated in this database are set on the hg19 genome assembly coordinates. We annotated all the nucleotide substitutions with a frequency greater than 10% and a coverage of a minimum of 10 counts in both coding, noncoding, and regulatory sequences (i.e., 5′ and 3′ untranslated and flanking upstream and downstream regulatory regions) for each gene of interest to consider the possible damage and fragmentation to which the ancient hominin DNA was subjected. Of note, the variant frequency indicated the percentage of frequency of that substitution in ancient hominines, as reported by the IGV tool, considering the depth (coverage) of the reads displayed at each locus. For each tested gene, a region of approximately 500 bp upstream and downstream of the first and last exons was, respectively, considered and screened to annotate nucleotide substitutions within regulatory regions able to affect the gene expression rate. The precise hg19 genomic coordinates for each tested gene locus were as follows: *PRB1* locus 11,509,000–11,504,200 on chromosome 12; *PRB2* locus 11,549,000–11,544,000 on chromosome 12; *PRB3* locus 11,423,140–11,418,300 on chromosome 12; *PRB4* locus 11,463,900–11,459,500 on chromosome 12; *PRH2* locus 11,081,500–11,087,950 on chromosome 12; *HTN1* locus 70,915,750–70,925,000 on chromosome 4; *HTN3* locus 70,893,670–70,902,700 on chromosome 4; *AMY1A* locus 104,239,500–104,229,500 on chromosome 1; *STATH* locus 70,861,200–70,868,790 on chromosome 4; *SMR3B* locus 71,248,550–71,256,400 on chromosome 4; *CST1* locus 23,732,000–23,727,600 on chromosome 20; *CST2* locus 23,807,800–23,803,900 on chromosome 20; *CST3* locus 23,619,100–23,606,800 on chromosome 20; *CST4* locus 23,670,200–23,665,700 on chromosome 20; *CST5* locus 23,860,900–23,856,000 on chromosome 20; *CSTA* locus 122,043,600–122,061,300 on chromosome 3; and *CSTB* locus 45,196,800–45,193,000 on chromosome 21.

The annotation with the corresponding frequency of all variations in present-day human populations was collected by integrating information from both the dbSNP (Single Nucleotide Polymorphism Database; https://www.ncbi.nlm.nih.gov/snp, accessed on 15 July 2020) and the Ensembl (http://www.ensembl.org/index.html, accessed on 15 July 2020) databases. In particular, the frequency was reported as the Allele Frequency Aggregator (ALFA New). The analysis of regulatory regions in the gene loci analysed was assessed by implementing the information available on the UCSC Genome Browser database (https://genome.ucsc.edu, accessed on 15 July 2020).

The coding sequences of salivary proteins were extracted from the publicly available UniProtKB database (https://www.uniprot.org/, accessed on 15 July 2020): PRB1, primary accession number: P04280; PRB2: P02812; PRB3: Q04118; PRB4: P10163; PRH2: P02810; HTN1: P15515; HTN3: P15516; STATH: P02808; AMY1A: P0DUB6; P-B: P02814, CST1: P01037; CST2: P09228; CST3: P01034; CST4: P01036; CST5: P28325, CSTA: P01040, CSTB: P04080.

### 4.2. Protein Data Analysis

The potential impact of the amino acid substitution on salivary protein function was predicted by SIFT (sorting intolerant from tolerant) version 5.1.1 using the Genome tool (SIFT nonsynonymous single nucleotide variants (genome-scale), available at the SIFT website (http://sift.jcvi.org/, accessed on 20 June 2022). The SIFT algorithm is based on the degree of conservation of amino acid residues in sequence alignments derived from closely related sequences, collected through PSI-BLAST [62]. SIFT results with a score < 0.05 indicate amino acids deleterious on protein function.

### 4.3. Selective Pressure Analysis

To detect any possible trace of selective pressure, PBS has been applied. PBS is a statistical three-population test based on the FST fixation index, and it has proven to be one of the best methods of detecting signs of recent natural selection on genomes [31]. Regarding the choice of the three populations, we used three distant populations worldwide (CEU for Europe, CHB for Asia, and YRI for Africa), which are the most commonly used [63,64] and are among the first populations released by the 1000 Genomes, Phase 1 [64].

FST among three possible populations pairs (CEU, CHB, and YRI) has been calculated by VCFtools v0.1.16 [65] using VCF files of each gene under scrutiny. The genes were previously filtrated with Plink 1.9 [66] to keep only the variants with MAF ≥ 0.05. Then, PBS and relative plots were performed with R Studio software (R Core Team 2021, https://www.R-project.org, accessed on 2 December 2022).

## 5. Conclusions

In conclusion, the nucleotide substitutions that have putatively affected the amino acid composition, the post-translational modification, and/or the gene expression levels of salivary proteins described in this study might have generated novel functional features and a different expression ratio among the several components of the salivary proteome. Given the largely unknown functional roles of most salivary proteins, we may only speculate that these changes could have ultimately modified the entire homeostasis of the oral cavity environment, possibly conditioning the eating habit lifestyle of modern humans. Our data may pave the way to unravelling evolutionary processes that have occurred through changes of salivary composition in the oral cavity homeostasis. This knowledge could provide additional novel cues toward a better understanding of the ability of different species to adapt to different and changing environments.

## Figures and Tables

**Figure 1 ijms-24-15010-f001:**
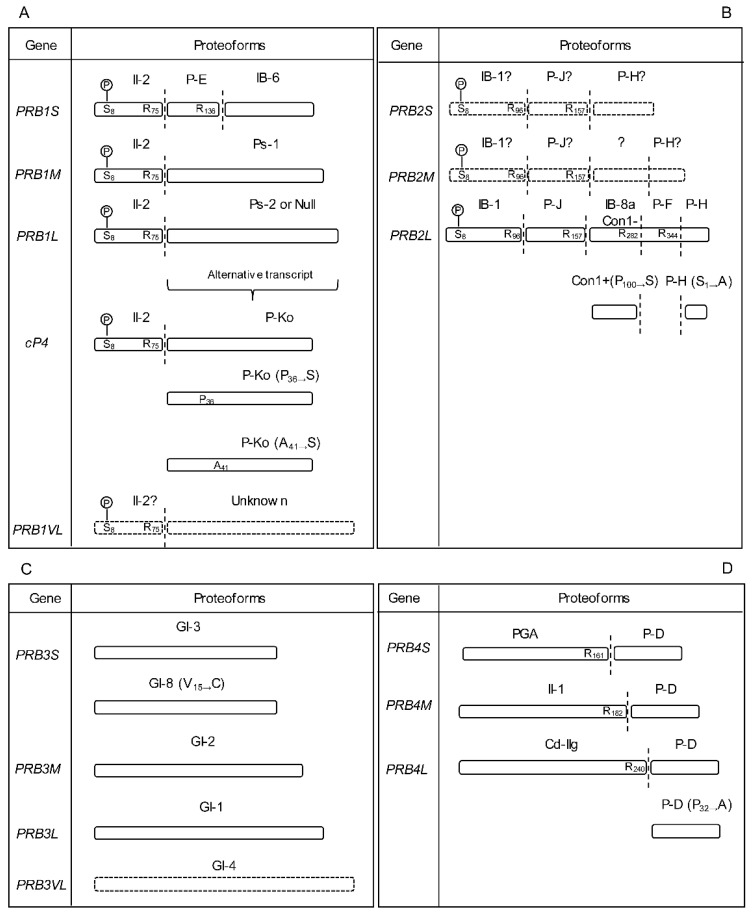
Schematic representation of basic proline-rich genes and encoded proteins: PRB1 (**A**), PRB2 (**B**), PRB3 (**C**), PRB4 (**D**). For each protein, the genetic allelic variants (S, small; M, medium; L, large; and VL, very large) are shown on the left-sided column; the resulting alternative proteoforms are shown on the right-sided column as blocks, with the corresponding symbol on top. Vertical dashed lines indicate the pro-protein convertase cleavage sites with corresponding Arg (R) residues’ positions. The P enclosed in a circle denotes phosphorylation sites; aminoacidic substitutions are shown for selected isoforms. See text for additional details.

**Figure 2 ijms-24-15010-f002:**
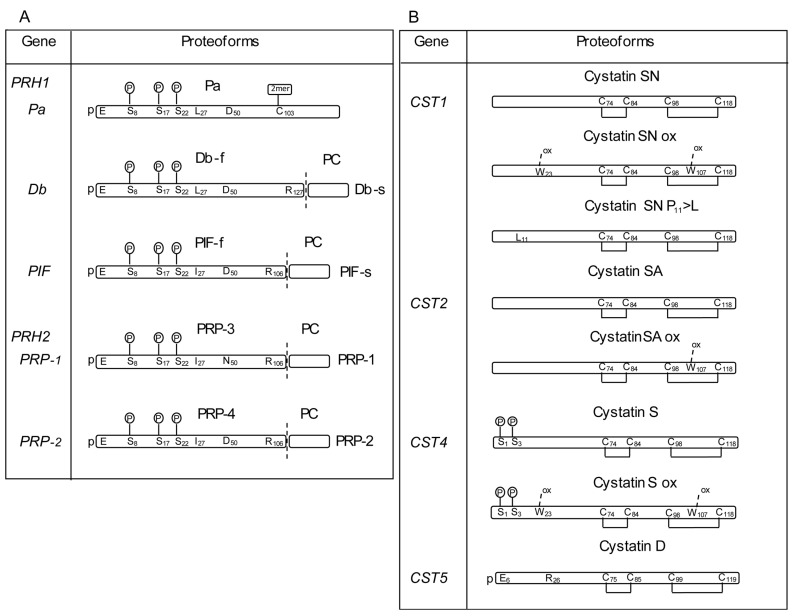
Schematic representation of acidic proline-rich proteins (**A**) and cystatins (**B**). For each protein, the genetic allelic variants (S, small; M, medium; L, large; and VL, very large) are shown on the left-sided column; the resulting alternative proteoforms are shown on the right-sided column as blocks with corresponding symbols on top. All cystatin alternative proteoforms feature two disulfide bridges (indicated by brackets between Cys), oxidation (ox), and phosphorylation (P) sites. Vertical dashed lines indicate the pro-protein convertase cleavage sites with corresponding Arg (R) residues’ positions. The P enclosed in a circle denotes phosphorylation sites; ox: oxidation sites; p-E: N-terminal pyroglutamic acid; aminoacidic substitutions are shown for selected isoforms. See text for additional details.

**Figure 3 ijms-24-15010-f003:**
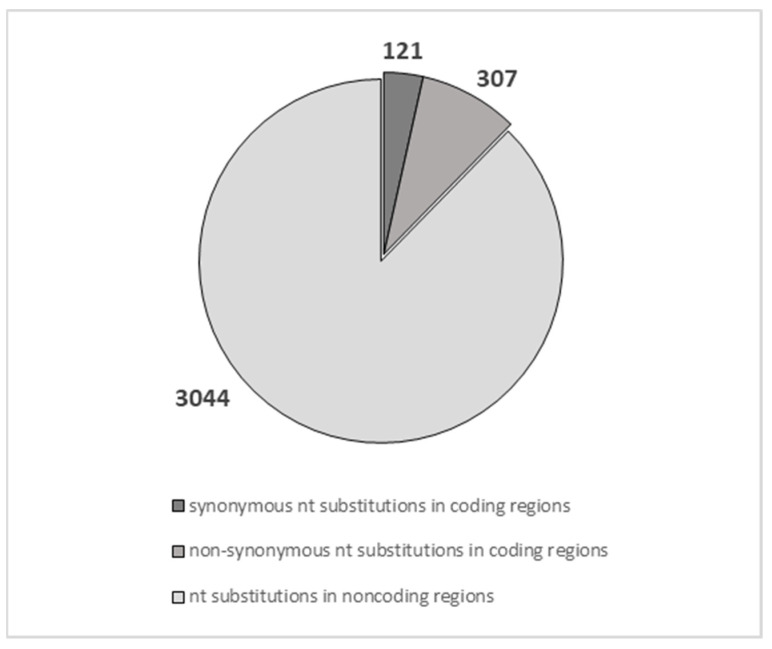
Nucleotide substitutions in salivary protein genes. The pie chart shows the type and number of 3472 nucleotide substitutions across the 17 tested salivary genes. In particular, the 428 substitutions found in coding regions included 307 nonsynonymous changes across all the 17 genes tested. See text for additional details.

**Figure 4 ijms-24-15010-f004:**
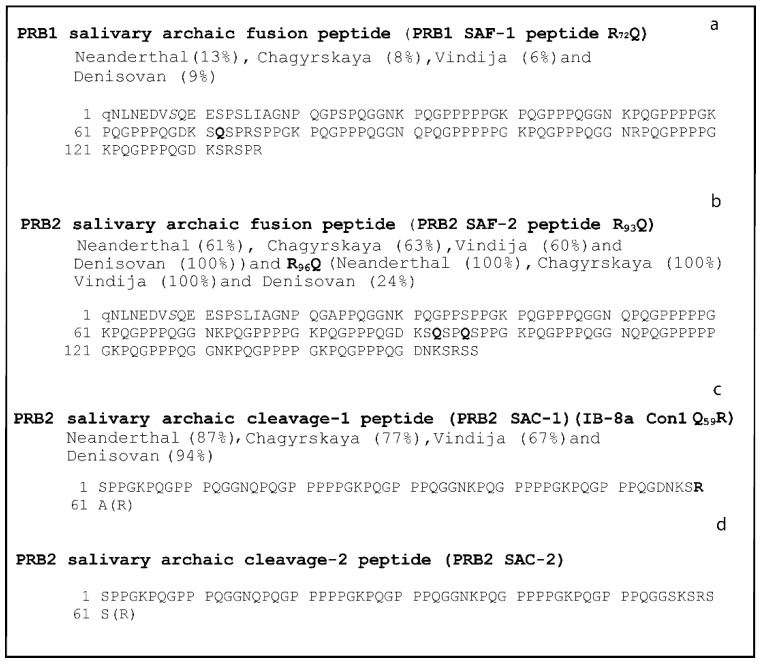
Predicted archaic hominins’ PRB-1 (panel (**a**)) and PRB-2 (panels (**b**–**d**)) protein variants.

**Table 1 ijms-24-15010-t001:** Neanderthal and Denisovan nucleotide substitutions and the corresponding SIFT results on *PRB1*, *PRB2*, *PRB3*, and *PRB4* gene loci.

Chromosome Position (hg19)	Gene Region	Modern Human	Altai Neanderthal (Variant Frequency ^a^)	Chagyrskaya Neanderthal (Variant Frequency ^a^)	Vindija Neanderthal (Variant Frequency ^a^)	Denisovan (Variant Frequency ^a^)	Codon→Amino Acid	SNP id	SNP Total Frequency (ALFA)	SIFT Results (Score)
***PRB1* (reverse reading, chromosome 12)**										
**11,507,477**	Exon 2 (II-2)	**C**TT	**C**TT (100%)	**T**TT (13%)	**T**TT (7%) *	**C**TT (100%)	GAA→E_10_ AAA→K_10_	n.a.	n.a.	Damaging (0.02)
**11,507,464**	Exon 2 (II-2)	A**G**G	A**G**G (100%)	A**G**G (100%)	A**A**G (12%)	A**G**G (100%)	UCC→S_14_ UUC→F_14_	rs1173856027	A = 0%	Tolerated (0.72)
**11,506,888**	Exon 3 (II-2)	G**G**G	G**G**G (100%)	G**G**G (100%)	G**A**G (12%)	G**G**G (100%)	CCC→P_35_ CUC→L_35_	n.a.	n.a.	Tolerated (0.06)
**11,506,856**	Exon 3 (II-2)	**G**GG	**G**GG (100%)	**A**GG (11%)	**G**GG (100%)	**G**GG (100%)	CCC→P_45_ UCC→S_45_	rs762910991	A = 0.003%	Tolerated (0.17)
**11,506,853**	Exon 3 (II-2)	**G**GT	**T**GT (3%) *	**G**GT (100%)	**A**GT (15%)	**G**GT (100%)	CCA→P_46_ UCA→S_46_	rs745726339	A = 0%	Damaging (0)
**11,506,852**	Exon 3 (II-2)	G**G**T	G**G**T (100%)	G**G**T (100%)	G**A**T (11%)	G**G**T (100%)	CCA→P_46_ CUA→L_46_	n.a.	n.a.	Damaging (0)
**11,506,804**	Exon 3 (II-2)	G**T**T	G**A**T (61%)	G**A**T (63%)	G**A**T (60%)	G**T**T (100%)	CAA→Q_62_ CUA→L_62_	n.a.	n.a.	Tolerated (0.29)
**11,506,801**	Exon 3 (II-2)	C**C**T	C**C**T (100%)	C**T**T (11%)	C**T**T (5%) *	C**C**T (100%)	GGA→G_63_ GAA→E_63_	n.a.	n.a.	Damaging (0.01)
**11,506,790**	Exon 3 (II-2)	**G**TT	**G**TT (100%)	**A**TT (11%)	**A**TT (6%) *	**G**TT (100%)	CAA→Q_67_ UAA→stop	rs1409612167	A = 0%	Damaging due to stop
**11,506,784**	Exon 3 (II-2)	**C**TG	**C**TG (100%)	**C**TG (100%)	**T**TG (13%)	**C**TG (100%)	GAC→D_69_ AAC→N_69_	rs554211998	T = 0%	Tolerated (0.95)
**11,506,774**	Exon 3 (II-2)	G**C**T	G**T**T (13%)	G**T**T (8%) *	G**T**T (6%) *	G**T**T (9%) *	CGA→R_72_ CAA→Q_72_	rs202083397	T = 10.6%	Tolerated (0.08)
**11,506,766**	Exon 3 (II-2)	**G**CT	**G**CT (100%)	**G**CT (100%)	**A**CT (12%)	**G**CT (100%)	CGA→R_75_ UGA→stop	rs766131639	A = 0%	Damaging due to stop
**11,506,730**	Exon 3 (Ps-2)	**G**TT	**G**TT (100%)	**A**TT (16%)	**G**TT (100%)	**G**TT (100%)	CAA→Q_12_ UAA→stop	n.a.	n.a.	Damaging due to stop
**11,506,723**	Exon 3 (Ps-2)	C**C**A	C**C**A (100%)	C**T**A (12%)	C**T**A (3%) *	C**C**A (100%)	GGU→G_14_ GAU→D_14_	rs534597111	T = 0%	NS
**11,506,669**	Exon 3 (Ps-2)	G**G**T	G**T**T (39%)	G**T**T (36%)	G**T**T (55%)	G**T**T (26%)	CCA→P_32_ CAA→Q_32_	rs772365043	C = 0%	NS
**11,506,618**	Exon 3 (Ps-2)	C**C**T	C**C**T (100%)	C**T**T (17%)	C**T**T (3%) *	C**C**T (100%)	GGA→G_49_ GAA→E_49_	n.a.	n.a.	NS
**11,506,612**	Exon 3 (Ps-2)	G**G**G	G**G**G (100%)	G**A**G (11%)	G**G**G (100%)	G**G**G (100%)	CCC→P_51_ CUC→L_51_	n.a.	n.a.	NS
**11,506,577**	Exon 3 (IB-6)	**G**GA	**G**GA (100%)	**A**GA (13%)	**G**GA (100%)	**G**GA (100%)	CCU→P_2_ UCU→S_2_	n.a.	n.a.	NS
**11,506,514**	Exon 3 (IB-6)	**G**GA	**G**GA (100%)	**A**GA (6%) *	**A**GA (11%)	**G**GA (100%)	CCU→P_23_ UCU→S_23_	n.a.	n.a.	NS
**11,506,492**	Exon 3 (IB-6)	G**G**T	G**G**T (100%)	G**G**T (100%)	G**A**T (13%)	G**G**T (100%)	CCA→P_30_ CUA→L_30_	n.a.	n.a.	NS
**11,506,490**	Exon 3 (IB-6)	**G**GG	**A**GG (5%) *	**A**GG (18%)	**A**GG (8%) *	**G**GG (100%)	CCC→P_31_ UCC→S_31_	n.a.	n.a.	NS
**11,506,486**	Exon 3 (IB-6)	G**G**T	G**G**T (100%)	G**G**T (100%)	G**T**T (18%)	G**G**T (100%)	CCA→P_32_ CAA→Q_32_	rs755622101	T = 1.3%	NS
**11,506,473**	Exon 3 (Ps-2)	TT**C**	TT**G**(100%)	TT**G**(83%) **	TT**G**(100%) **	TT**G**(75%) **	AAG→K_37_ AAC→N_37_	rs61930109	G = 72.1%	NS
**11,506,403**	Exon 3 (Ps-2)	**A**GG	**G**GG (50%) **	**G**GG (50%) **	**A**GG (100%) **	**G**GG (100%)	UCC→S_59_ CCC→P_59_	n.a.	n.a.	NS
**11,506,370**	Exon 3 (Ps-2)	**G**GG	**G**GG (100%)	**G**GG (100%)	**A**GG (21%)	**G**GG (100%)	CCC→P_70_ UCC→S_70_	rs774158904	A = 0%	NS
**11,506,369**	Exon 3 (Ps-2)	G**G**G	G**G**G (93%)	G**G**G (100%)	G**A**G (16%)	G**G**G (100%)	CCC→P_71_ CUC→L_71_	rs369001998	A = 0.007%	NS
**11,506,339**	Exon 3 (Ps-2)	G**G**G	G**G**G (97%)	G**A**G (5%) *	G**A**G (23%)	G**G**G (100%)	CCC→P_81_ CUC→L_81_	n.a.	n.a.	NS
**11,506,333**	Exon 3 (Ps-2)	G**G**A	G**G**A (100%)	G**A**A (5%) *	G**A**A (11%)	G**G**A (100%)	CCU→P_83_ CUU→L_83_	n.a.	n.a.	NS
**11,506,309**	Exon 3 (Ps-2)	G**G**T	G**A**T (4%) *	G**A**T (6%) *	G**A**T (17%)	G**G**T (100%)	CCA→P_91_ CUU→L_91_	n.a.	n.a.	Damaging (0.01)
**11,506,303**	Exon 3 (Ps-2)	G**G**T	G**T**T (3%) *	G**T**T (13%)	G**G**T (100%)	G**G**T (100%)	CCA→P_93_ CAA→Q_93_	rs201682460	T = 2.8%	Damaging (0)
**11,506,301**	Exon 3 (Ps-2)	**G**TT	**A**TT (4%) *	**G**TT (100%)	**A**TT (15%)	**G**TT (100%)	CAA→Q_94_ UAA→stop	n.a.	n.a.	Damaging due to stop
**11,506,285**	Exon 3 (Ps-2)	G**G**A	G**G**A (100%)	G**G**A (100%)	G**A**A (14%)	G**G**A (100%)	CCU→P_99_ CUU→L_99_	n.a.	n.a.	Damaging (0.01)
**11,506,283**	Exon 3 (Ps-2)	**G**TT	**G**TT (100%)	**A**TT (14%)	**A**TT (13%)	**G**TT (100%)	CAA→Q_100_ UAA→stop	n.a.	n.a.	Damaging due to stop
**11,506,250**	Exon 3 (Ps-2)	**G**GT	**G**GT (100%) **	**G**GT (100%)	**A**GT (14%)	**G**GT (100%)	CCA→P_111_ UCA→S_111_	n.a.	n.a.	Tolerated (0.08)
**11,506,249**	Exon 3 (Ps-2)	G**G**T	G**G**T (100%) **	G**G**T (100%)	G**A**T (13%)	G**G**T (100%)	CCA→P_111_ CUA→L_111_	rs1208300501	A = 0%	Tolerated (0.09)
**11,506,246**	Exon 3 (Ps-2)	G**G**G	G**G**G (100%) **	G**A**G (18%)	G**G**G (100%)	G**G**G (100%)	CCC→P_112_ CUC→L_112_	rs1303924609	A = 0%	Damaging (0.02)
**11,506,241**	Exon 3 (Ps-2)	**G**TT	**G**TT (100%) **	**G**TT (100%)	**A**TT (14%)	**G**TT (100%)	CAA→Q_114_ UAA→stop	rs751826141	A = 0%	Damaging due to stop
**11,506,217**	Exon 3 (IB-6)	**C**GG	**G**GG (67%) **	**G**GG (17%) **	**G**GG (25%)	**C**GG (100%)	GCC→A_61_ CCC→P_61_	rs771648794	G = 0.04%	Tolerated (1)
**11,506,154**	Exon 3 (IB-6)	**G**GG	**G**GG (100%)	**A**GG (17%)	**A**GG (4%) *	**G**GG (100%)	CCC→P_82_ UCC→S_82_	n.a.	n.a.	Tolerated (0.15)
**11,506,150**	Exon 3 (IB-6)	G**G**T	G**G**T (100%)	G**A**T (14%)	G**G**T (100%)	G**A**T (6%) *	CCA→P_83_ CUA→L_83_	rs747444571	A = 0%	Damaging (0.03)
**11,506,079**	Exon 3 (IB-6)	**G**GA	**G**GA (100%)	**G**GA (100%)	**A**GA (13%)	**G**GA (100%)	CCU→P_107_ UCU→S_107_	n.a.	n.a.	Tolerated (0.06)
**11,506,075**	Exon 3 (IB-6)	G**G**A	G**G**A (100%)	G**G**A (100%)	G**A**A (13%)	G**G**A (100%)	CCU→P_108_ CUU→L_108_	n.a.	n.a.	Damaging (0.01)
**11,506,070**	Exon 3 (IB-6)	**C**CC	**C**CC (100%)	**C**CC (100%)	**T**CC (12%)	**C**CC (100%)	GGG→G_110_ AGG→R_110_	n.a.	n.a.	Tolerated (0.3)
**11,506,057**	Exon 3 (IB-6)	A**G**G	A**G**G (100%)	A**A**G (11%)	A**A**G (5%) *	A**G**G (100%)	UCC→S_114_ UUC→F_114_	n.a.	n.a.	Damaging (0.03)
**11,506,052**	Exon 3 (IB-6)	**G**GA	**G**GA (100%)	**A**GA (10%) *	**A**GA (18%)	**G**GA (100%)	CCU→P_116_ UCU→S_116_	rs1372423355	A = 0%	Tolerated (0.06)
***PRB2* (reverse reading, chromosome 12)**										
**11,548,429**	Exon 1 (Signal)	C**G**G	C**G**G (100%)	C**A**G (3%) *	C**A**G (13%)	C**G**G (100%)	GCC→A_11(sp)_ GUC→V_11(sp)_	rs1415819382	A = 0%	Damaging (0)
**11,547,429**	Exon 2 (IB-1)	**C**CT	**T**CT (4%) *	**C**CT (100%)	**T**CT (12%)	**C**CT (100%)	GGA→G_18_ AGA→R_18_	n.a.	n.a.	Damaging (0.2)
**11,546,899**	Exon 3 (IB-1)	C**C**T	C**C**T (100%)	C**T**T (11%)	C**C**T (100%)	C**C**T (100%)	GGA→G_22_ GAA→E_22_	rs188924826	T = 0.007%	Tolerated (0.1)
**11,546,894**	Exon 3 (IB-1)	**G**GG	**G**GG (100%)	**A**GG (14%)	**G**GG (100%)	**G**GG (100%)	CCC→P_24_ UCC→S_24_	n.a.	n.a.	Tolerated (0.73)
**11,546,872**	Exon 3 (IB-1)	G**G**A	G**G**A (100%)	G**G**A (100%)	G**A**A (11%)	G**G**A (100%)	CCU→P_31_ CUU→L_31_	rs748769813	A = 0%	Tolerated (0.46)
**11,546,830**	Exon 3 (IB-1)	G**G**G	G**G**G (100%)	G**A**G (9%) *	G**A**G (17%)	G**G**G (100%)	CCC→P_45_ CUC→L_45_	n.a.	n.a.	Tolerated (0.1)
**11,546,828**	Exon 3 (IB-1)	**G**GT	**A**GT (3%) *	**G**GT (100%)	**A**GT (17%)	**G**GT (100%)	CCA→P_46_ UCA→S_46_	rs755161117	A = 0.007%	Tolerated (0.36)
**11,546,825**	Exon 3 (IB-1)	**G**TT	**G**TT (97%)	**G**TT (100%)	**A**TT (17%)	**G**TT (100%)	CAA→Q_47_ UAA→stop	n.a.	n.a.	Damaging due to stop
**11,546,810**	Exon 3 (IB-1)	**G**GA	**G**GA (100%)	**G**GA (100%)	**A**GA (13%)	**G**GA (100%)	CCU→P_52_ UCU→S_52_	rs1347881375	A = 0%	Tolerated (0.97)
**11,546,809**	Exon 3 (IB-1)	G**G**A	G**G**A (100%)	G**A**A (6%) *	G**A**A (12%)	G**G**A (100%)	CCU→P_52_ CUU→L_52_	n.a.	n.a.	Tolerated (0.3)
**11,546,807**	Exon 3 (IB-1)	**G**TT	**G**TT (97%)	**A**TT (11%)	**A**TT (11%)	**G**TT (100%)	CAA→Q_53_ UAA→stop	n.a.	n.a.	Damaging due to stop
**11,546,792**	Exon 3 (IB-1)	**G**GA	**G**GA (100%)	**A**GA (18%)	**G**GA (100%)	**G**GA (100%)	CCU→P_58_ UCU→S_58_	n.a.	n.a.	Tolerated (0.76)
**11,546,780**	Exon 3 (IB-1)	**G**GT	**G**GT (100%)	**G**GT (100%)	**A**GT (12%)	**G**GT (100%)	CCA→P_62_ UCA→S_62_	n.a.	n.a.	Tolerated (0.64)
**11,546,770**	Exon 3 (IB-1)	G**G**T	G**G**T (100%)	G**G**T (100%)	G**A**T (13%)	G**G**T (100%)	CCA→P_65_ CUA→L_65_	n.a.	n.a.	Tolerated (1)
**11,546,764**	Exon 3 (IB-1)	G**G**T	G**G**T (100%)	G**G**T (96%)	G**A**T (12%)	G**G**T (100%)	CCA→P_67_ CAA→Q_67_	rs201994479	T = 0.008%	Tolerated (0.43)
**11,546,732**	Exon 3 (IB-1)	**G**GA	**G**GA (100%)	**G**GA (100%)	**A**GA (13%)	**G**GA (100%)	CCU→P_78_ UCU→S_78_	n.a.	n.a.	Tolerated (0.38)
**11,546,716**	Exon 3 (IB-1)	G**T**T	G**A**T (4%) *	G**A**T (14%)	G**T**T (97%)	G**T**T (100%)	CAA→Q_83_ CUA→L_83_	n.a.	n.a.	Tolerated (0.32)
**11,546,686**	Exon 3 (IB-1)	G**C**T	G**T**T (42%)	G**T**T (39%)	G**T**T (51%)	G**T**T (29%)	CGA→R_93_ CAA→Q_93_	rs76832300	n.a.	Tolerated (0.5)
**11,546,677**	Exon 3 (IB-1)	G**C**T	G**C**T (100%)	G**C**T (100%)	G**C**T (100%)	G**T**T (24%)	CGA→R_96_ CAA→Q_96_	rs201144571	T = 0.08%	Tolerated (0.47)
**11,546,647**	Exon 3 (P-J)	G**G**G	G**G**G (100%)	G**G**G (100%)	G**A**G (15%)	G**G**G (100%)	CCC→P_10_ CUC→L_10_	n.a.	n.a.	Tolerated (0.18)
**11,546,642**	Exon 3 (P-J)	**G**TT	**G**TT (100%)	**G**TT (100%)	**A**TT (17%)	**G**TT (100%)	CAA→Q_12_ UAA→stop	n.a.	n.a.	Damaging due to stop
**11,546,627**	Exon 3 (P-J)	**G**GA	**A**GA (3%) *	**A**GA (11%)	**A**GA (5%) *	**G**GA (100%)	CCU→P_17_ UCU→S_17_	n.a.	n.a.	Tolerated (0.45)
**11,546,618**	Exon 3 (P-J)	**G**GA	**G**GA (100%)	**G**GA (93%)	**A**GA (17%)	**G**GA (100%)	CCU→P_20_ UCU→S_20_	n.a.	n.a.	Tolerated (0.81)
**11,546,617**	Exon 3 (P-J)	G**G**A	G**G**A (100%)	G**G**A (100%)	G**A**A (17%)	G**G**A (100%)	CCU→P_20_ CUU→L_20_	rs780517289	A = 0%	Tolerated (0.82)
**11,546,615**	Exon 3 (P-J)	**G**GT	**G**GT (100%)	**A**GT (12%)	**A**GT (8%) *	**G**GT (100%)	CCA→P_21_ UCA→S_21_	n.a.	n.a.	Tolerated (0.39)
**11,546,614**	Exon 3 (P-J)	G**G**T	G**G**T (100%)	G**A**T (11%)	G**G**T (100%)	G**G**T (100%)	CCA→P_21_ CUA→L_21_	n.a.	n.a.	Tolerated (0.29)
**11,546,585**	Exon 3 (P-J)	**G**GG	**G**GG (100%)	**G**GG (100%)	**A**GG (13%)	**G**GG (100%)	CCC→P_31_ UCC→S_31_	n.a.	n.a.	Tolerated (0.53)
**11,546,581**	Exon 3 (P-J)	G**G**T	G**T**T (6%) *	G**T**T (13%)	G**G**T (100%)	G**G**T (100%)	CCA→ P_32_ CAA→Q_32_	n.a.	n.a.	Damaging (0.05)
**11,546,566**	Exon 3 (P-J)	T**T**T	T**C**T (8%) *	T**C**T (12%)	T**T**T (100%)	T**T**T (100%)	AAA→K_37_ AGA→R_37_	rs746515947	C = 0%	Tolerated (1)
**11,546,462**	Exon 3 (IB-8a)	**G**GG	**G**GG (100%)	**A**GG (13%)	**G**GG (100%)	**G**GG (100%)	CCC→P_9_ UCC→S_9_	rs201392419	A = 0%	Tolerated (0.58)
**11,546,395**	Exon 3 (IB-8a)	G**G**T	G**T**T (16%)	G**T**T (10%) *	G**T**T (13%)	G**T**T (4%) *	CCA→P_31_ CAA→Q_31_	rs11054277	T = 0.01%	Damaging (0)
**11,546,380**	Exon 3 (IB-8a)	T**T**T	T**C**T (17%)	T**C**T (14%)	T**C**T (6%) *	T**T**T (100%)	AAA→K_37_ AGA→R_37_	rs11054276	C = 0.01%	Tolerated (1)
**11,546,381**	Exon 3 (IB-8a)	**T**TT	**T**TT (100%)	**C**TT (100%)	**T**TT (100%)	**G**TT (13%)	AAA→K_37_ CAA→Q_37_	rs201455726	G = 0.2%	Tolerated (0.42)
**11,546,369**	Exon 3 (IB-8a)	**G**GG	**G**GG (100%)	**A**GG (12%)	**G**GG (100%)	**G**GG (100%)	CCC→P_41_ UCC→S_41_	rs1238238576	A = 0%	Tolerated (0.42)
**11,546,347**	Exon 3 (IB-8a)	G**T**T	G**A**T (6%) *	G**A**T (4%) *	G**A**T (15%)	G**T**T (100%)	CAA→Q_48_ CUA→L_48_	n.a.	n.a.	Tolerated (0.32)
**11,546,342**	Exon 3 (IB-8a)	**G**GT	**G**GT (100%)	**G**GT (100%)	**A**GT (18%)	**G**GT (100%)	CCA→P_50_ UCA→S_50_	n.a.	n.a.	Tolerated (0.41)
**11,546,327**	Exon 3 (IB-8a)	**C**TG	**C**TG (100%)	**T**TG (11%)	**T**TG (18%)	**C**TG (100%)	GAC→D_55_ AAC→N_55_	n.a.	n.a.	Tolerated (0.28)
**11,546,314**	Exon 3 (IB-8a)	G**T**T	G**C**T (87%)	G**C**T (77%)	G**C**T (67%)	G**C**T (94%)	CAA→Q_59_ CGA→R_59_	rs34305575	C = 7.6%	Tolerated (0.35)
**11,546,309**	Exon 3 (IB-8a)	**C**GG	**G**GG (12%)	**G**GG (13%)	**G**GG (18%)	**G**GG (5%) *	GCC→A_61_ CCC→P_61_	rs201308939	G = 3.8%	Tolerated (0.25)
**11,546,305**	Exon 3 (IB-8a)	G**C**T	G**T**T (3%) *	G**C**T (100%)	G**T**T (11%)	G**C**T (100%)	CGA→R_62_ CAA→Q_62_	rs199748368	T = 0.07%	Tolerated (0.46)
**11,546,300**	Exon 3 (IB-8a)	**G**GA	**G**GA (100%)	**A**GA (13%)	**G**GA (100%)	**G**GA (100%)	CCU→P_64_ UCU→S_64_	rs755713521	n.a.	Tolerated (0.66)
**11,546,294**	Exon 3 (IB-8a)	**C**CT	**C**CT (100%)	**T**CT (13%)	**C**CT (100%)	**C**CT (100%)	GGA→G_66_ AGA→R_66_	n.a.	n.a.	Damaging (0.03)
**11,546,279**	Exon 3 (IB-8a)	**G**GT	**A**GT (2%) *	**G**GT (100%)	**A**GT (13%)	**G**GT (100%)	CCA→P_71_ UCA→S_71_	n.a.	n.a.	Tolerated (0.67)
**11,546,278**	Exon 3 (IB-8a)	G**G**T	G**A**T (2%) *	G**G**T (100%)	G**A**T (13%)	G**G**T (100%)	CCA→P_71_ CUA→L_71_	rs766408532	n.a.	Tolerated (0.26)
**11,546,246**	Exon 3 (IB-8a)	**G**GG	**G**GG (100%)	**G**GG (100%)	**A**GG (14%)	**G**GG (100%)	CCC→P_82_ UCC→S_82_	rs1440556057	A = 0.0004%	Tolerated (0.42)
**11,546,245**	Exon 3 (IB-8a)	G**G**G	G**G**G (97%)	G**A**G (7%) *	G**A**G (26%)	G**A**G (7%) *	CCC→P_82_ CUC→L_82_	rs1262267049	A = 0.0004%	Tolerated (0.15)
**11,546,213**	Exon 3 (IB-8a)	**G**GG	**G**GG (100%)	**A**GG (8%) *	**A**GG (25%)	**G**GG (100%)	CCC→P_93_ UCC→S_93_	rs1408969762	n.a.	Tolerated (0.26)
**11,546,187**	Exon 3 (IB-8a)	GT**T**	GT**T** (96%)	GT**C** (10%) *	GT**C** (12%)	GT**C** (4%) *	CAA→Q_101_ CAC→H_101_	n.a.	n.a.	Tolerated (0.23)
**11,546,161**	Exon 3 (IB-8a)	G**T**T	G**A**T (21%)	G**T**T (100%)	G**A**T (30%)	G**T**T (100%)	CAA→Q_110_ CUA→L_110_	n.a.	n.a.	Tolerated (0.61)
**11,546,089**	Exon 3 (P-F)	G**G**G	G**G**G (100%)	G**A**G (17%) **	G**A**G (17%)	G**G**G (100%)	CCC→P_10_ CUC→L_10_	n.a.	n.a.	Tolerated (0.61)
**11,546,084**	Exon 3 (P-F)	**G**TT	**G**TT (100%)	**G**TT (100%)	**A**TT (15%)	**G**TT (100%)	CAA→Q_12_ UAA→stop	n.a.	n.a.	Damaging due to stop
**11,546,059**	Exon 3 (P-F)	G**G**G	G**G**G (100%)	G**A**G (7%) *	G**A**G (21%)	G**G**G (100%)	CCC→P_20_ CUC→L_20_	n.a.	n.a.	Tolerated (0.19)
**11,546,050**	Exon 3 (P-F)	G**G**A	G**T**A (4%) *	G**T**A (13%)	G**G**A (100%)	G**T**A (7%) *	CCU→P_23_ CAU→H_23_	n.a.	n.a.	Tolerated (0.56)
**11,546,027**	Exon 3 (P-F)	**G**GG	**G**GG (100%)	**A**GG (11%)	**A**GG (7%) *	**G**GG (100%)	CCC→P_31_ UCC→S_31_	rs1201001162	n.a.	Tolerated (0.61)
**11,546,023**	Exon 3 (P-F)	G**G**T	G**G**T (100%)	G**T**T (5%) *	G**T**T (13%)	G**T**T (4%) *	CCA→P_32_ CAA→Q_32_	rs201391404	T = 0.059%	Damaging (0.03)
**11,546,009**	Exon 3 (P-F)	**T**TT	**T**TT (100%)	**T**TT (100%)	**T**TT (95%)	**G**TT (12%)	AAA→K_37_ CAA→ Q_37_	n.a.	n.a.	Tolerated (0.26)
**11,545,975**	Exon 3 (P-F)	G**T**T	G**A**T (2%) *	G**A**T (16%)	G**A**T (33%)	G**T**T (100%)	CAA→Q_48_ CUA→L_48_	n.a.	n.a.	Tolerated (0.31)
**11,545,964**	Exon 3 (P-F)	**G**GT	**G**GT (100%)	**C**GT (20%)	**C**GT (22%)	**C**GT (19%)	CCA→P_51_ GCA→A_51_	n.a.	n.a.	Tolerated (0.74)
**11,545,904**	Exon 3 (P-H)	**G**GG	**G**GG (100%)	**A**GG (3%) *	**A**GG (11%)	**G**GG (100%)	CCC→P_10_ UCC→S_10_	n.a.	n.a.	Tolerated (0.8)
**11,545,868**	Exon 3 (P-H)	**G**GA	**G**GA (100%)	**G**GA (100%)	**A**GA (13%)	**G**GA (100%)	CCU→P_22_ UCU→S_22_	n.a.	n.a.	Tolerated (0.69)
**11,545,814**	Exon 3 (P-H)	**G**TC	**G**TC (100%)	**A**TC (4%) *	**A**TC (12%)	**G**TC (100%)	CAG→Q_40_ UAG→stop	n.a.	n.a.	Damaging due to stop
**11,545,802**	Exon 3 (P-H)	**G**CG	**G**CG (100%)	**G**CG (100%)	**A**CG (11%)	**G**CG (100%)	CGC→R_44_ UGC→C_44_	rs748815572	A = 0%	Tolerated (0.07)
**11,545,793**	Exon 3 (P-H)	**G**TT	**G**TT (100%)	**A**TT (12%)	**G**TT (100%)	**G**TT (100%)	CAA→Q_47_ UAA→stop	n.a.	n.a.	Damaging due to stop
**11,545,790**	Exon 3 (P-H)	**C**CC	**C**CC (100%)	**C**CC (100%)	**T**CC (13%)	**C**CC (100%)	GGG→G_48_ AGG→R_48_	n.a.	n.a.	Tolerated (0.7)
***PRB3* (reverse reading, chromosome 12)**										
**11,422,578**	Exon 1 (Signal)	C**G**G	C**G**G (100%)	C**A**G (14%)	C**A**G (3%) *	C**G**G (100%)	GCC→A_8(sp)_ GUC→V_8(sp)_	rs1337927316	n.a.	Tolerated (0.06)
**11,421,578**	Exon 2 (Gl-5)	A**G**G	A**G**G (100%)	A**A**G (11%)	A**A**G (11%)	A**G**G (100%)	UCC→S_14_ UUC→F_14_	n.a.	n.a.	Tolerated (0.32)
**11,421,002**	Exon 3 (Gl-5)	**G**GG	**G**GG (100%)	**A**GG (11%)	**A**GG (4%) *	**G**GG (100%)	CCC→P_45_ UCC→S_45_	rs533382585	n.a.	Damaging (0.04)
**11,420,989**	Exon 3 (Gl-5)	C**C**G	C**C**G (100%)	C**T**G (14%)	C**T**G (5%) *	C**C**G (96%)	GGC→G_49_ GAC→D_49_	n.a.	n.a.	Damaging (0)
**11,420,975**	Exon 3 (Gl-5)	**C**CA	**T**CA (2%) *	**T**CA (17%)	**C**CA (100%)	**C**CA (100%)	GGU→G_54_ AGU→S_54_	rs1197023343	n.a.	Tolerated (0.12)
**11,420,974**	Exon 3 (Gl-5)	C**C**A	C**C**A (100%)	C**T**A (8%) *	C**T**A (21%)	C**C**A (100%)	GGU→G_54_ GAU→D_54_	n.a.	n.a.	Tolerated (0.19)
**11,420,971**	Exon 3 (Gl-5)	G**G**G	G**G**G (100%)	G**G**G (100%)	G**A**G (11%)	G**G**G (100%)	CCC→P_55_ CUC→L_55_	n.a.	n.a.	Damaging (0.02)
**11,420,956**	Exon 3 (Gl-5)	C**C**T	C**C**T (98%)	C**C**T (100%)	C**T**T (14%)	C**C**T (100%)	GGA→G_60_ GAA→E_60_	rs745804122	T = 0%	Tolerated (0.06)
**11,420,945**	Exon 3 (Gl-5)	**C**CT	**C**CT (100%)	**C**CT (100%)	**T**CT (14%)	**T**CT (4%) *	GGA→G_64_ AGA→R_64_	rs781151188	T = 0%	Damaging (0.02)
**11,420,939**	Exon 3 (Gl-5)	**G**GG	**G**GG (100%)	**A**GG (11%) **	**A**GG (11%)	**G**GG (100%)	CCC→P_66_ UCC→S_66_	n.a.	n.a.	Damaging (0.04)
**11,420,927**	Exon 3 (Gl-5)	**C**CT	**C**CT (100%)	**C**CT (100%)	**T**CT (11%)	**C**CT (100%)	GGA→G_70_ AGA→R_70_	n.a.	n.a.	Damaging (0)
**11,420,926**	Exon 3 (Gl-5)	C**C**T	C**C**T (100%)	C**C**T (100%)	C**T**T (16%)	C**C**T (100%)	GGA→G_70_ GAA→E_70_	n.a.	n.a.	Damaging (0)
**11,420,906**	Exon 3 (Gl-5)	**G**GT	**G**GT (100%)	**G**GT (100%)	**A**GT (12%)	**G**GT (100%)	CCA→P_77_ UCA→S_77_	n.a.	n.a.	Damaging (0.04)
**11,420,899**	Exon 3 (Gl-5)	G**C**A	G**T**A (73%)	G**C**A (100%)	G**T**A (65%)	G**T**A (80%)	CGU→R_79_ CAU→H_79_	rs769836435	T = 0.02%	Tolerated (0.59)
**11,420,896**	Exon 3 (Gl-5)	G**G**C	G**G**C (100%)	G**G**C (100%)	G**A**C (13%)	G**G**C (100%)	CCG→P_80_ CUG→L_80_	n.a.	n.a.	Tolerated (0.09)
**11,420,836**	Exon 3 (Gl-5)	G**C**A	G**T**A (7%) *	G**T**A (5%) *	G**T**A (9%) *	G**T**A (22%)	CGU→R_100_ CAU→H_100_	n.a.	n.a.	Tolerated (0.24)
**11,420,815**	Exon 3 (Gl-5)	G**G**T	G**T**T (18%)	G**G**T (100%)	G**G**T (96%)	G**G**T (100%)	CCA→P_107_ CAA→Q_107_	rs201963893	T = 0%	Tolerated (0.45)
**11,420,803**	Exon 3 (Gl-5)	C**C**T	C**C**T (100%)	C**C**T (100%)	C**T**T (15%)	C**C**T (100%)	GGA→G_111_ GAA→E_111_	n.a.	n.a.	Tolerated (0.41)
**11,420,800**	Exon 3 (Gl-5)	C**C**T	C**C**T (97%)	C**C**T (100%)	C**T**T (11%)	C**C**T (100%)	GGA→G_112_ GAA→E_112_	n.a.	n.a.	Damaging (0.01)
**11,420,780**	Exon 3 (Gl-5)	**G**GC	**G**GC (100%)	**A**GC (11%)	**G**GC (100%)	**G**GC (100%)	CCG→P_119_ UCG→S_119_	n.a.	n.a.	Damaging (0.04)
**11,420,779**	Exon 3 (Gl-5)	G**G**C	G**A**C (4%) *	G**A**C (6%) *	G**A**C (35%)	G**G**C (100%)	CCG→P_119_ CUG→L_119_	n.a.	n.a.	Damaging (0.03)
**11,420,728**	Exon 3 (Gl-5)	A**G**G	A**A**G (4%) *	A**G**G (100%)	A**A**G (11%)	A**G**G (100%)	UCC→S_136_ UUC→F_136_	n.a.	n.a.	Damaging (0.04)
**11,420,716**	Exon 3 (Gl-5)	G**G**C	G**A**C (4%) *	G**G**C (100%)	G**A**C (17%)	G**G**C (100%)	CCG→P_140_ CUG→L_140_	n.a.	n.a.	Tolerated (0.12)
**11,420,687**	Exon 3 (Gl-5)	**G**GG	**G**GG (98%)	**A**GG (15%)	**G**GG (100%)	**G**GG (100%)	CCC→P_150_ UCC→S_150_	n.a.	n.a.	Tolerated (0.15)
**11,420,686**	Exon 3 (Gl-5)	G**G**G	G**G**G (98%)	G**A**G (8%) *	G**A**G (18%)	G**G**G (100%)	CCC→P_150_ CUC→L_150_	n.a.	n.a.	Tolerated (0.15)
**11,420,614**	Exon 3 (Gl-2)	C**C**T	C**C**T (100%)	C**C**T (100%)	C**T**T (11%)	C**C**T (100%)	GGA→G_132_ GAA→E_132_	rs768625455	n.a.	NS
**11,420,597**	Exon 3 (Gl-2)	**C**CA	**C**CA (100%)	**C**CA (100%)	**T**CA (13%)	**C**CA (100%)	GGU→G_138_ AGU→S_138_	rs780713977	n.a.	Tolerated (0.09)
**11,420,588**	Exon 3 (Gl-2)	**G**GA	**A**GA (4%) *	**A**GA (10%) *	**A**GA (16%)	**G**GA (100%)	CCU→P_141_ UCU→S_141_	n.a.	n.a.	Tolerated (0.78)
**11,420,495**	Exon 3 (Gl-2)	**G**GT	**A**GT (12%)	**A**GT (3%) *	**A**GT (6%) *	**A**GT (14%)	CCA→P_172_ UCA→S_172_	n.a.	n.a.	Tolerated (0.14)
**11,420,308**	Exon 4 (Gl-2)	**G**GG	**G**GG (100%)	**A**GG (17%)	**G**GG (100%)	**G**GG (100%)	CCC→P_234_ UCC→S_234_	rs760324380	A = 0.0008%	Tolerated (0.09)
**11,420,307**	Exon 4 (Gl-2)	G**G**G	G**G**G (100%)	G**A**G (12%)	G**G**G (100%)	G**G**G (100%)	CCC→P_234_ CUC→L_234_	n.a.	n.a.	Damaging (0.03)
**11,420,304**	Exon 4 (Gl-2)	G**G**T	G**G**T (100%)	G**A**T (12%)	G**G**T (100%)	G**G**T (100%)	CCA→P_235_ CUA→L_235_	n.a.	n.a.	Damaging (0.01)
**11,420,281**	Exon 4 (Gl-2)	**G**CA	**G**CA (100%)	**A**CA (13%)	**A**CA (10%) *	**G**CA (100%)	CGU→R_243_ UGU→C_243_	rs758570507	A = 0%	Damaging (0.05)
**11,420,278**	Exon 4 (Gl-2)	**G**GG	**G**GG (100%)	**G**GG (100%)	**A**GG (11%)	**G**GG (100%)	CCC→P_244_ UCC→S_244_	n.a.	n.a.	Tolerated (0.27)
**11,420,182**	Exon 4 (Gl-2)	**G**GT	**G**GT (100%)	**G**GT (100%)	**A**GT (11%)	**G**GT (100%)	CCA→P_277_ UCA→S_277_	rs755939114	A = 0%	Tolerated (0.06)
**11,420,170**	Exon 4 (Gl-2)	**C**CC	**C**CC (100%)	**C**CC (100%)	**T**CC (11%)	**C**CC (100%)	GGG→G_280_ AGG→R_280_	n.a.	n.a.	Tolerated (0.07)
**11,420,161**	Exon 4 (Gl-2)	**G**GT	**G**GT (100%)	**G**GT (100%)	**A**GT (13%)	**G**GT (100%)	CCA→P_283_ UCA→S_283_	n.a.	n.a.	Tolerated (0.21)
**11,420,160**	Exon 4 (Gl-2)	G**G**T	G**G**T (100%)	G**G**T (100%)	G**A**T (19%)	G**G**T (100%)	CCA→P_283_ CUA→L_283_	n.a.	n.a.	Tolerated (0.09)
**11,420,154**	Exon 4 (Gl-2)	T**C**T	T**T**T (3%) *	T**C**T (100%)	T**T**T (11%)	T**C**T (100%)	AGA→R_285_ AAA→K_285_	n.a.	n.a.	Tolerated (0.63)
***PRB4* (reverse reading, chromosome 12)**										
**11,463,280**	Exon 1 (PGA)	T**C**A	T**G**A (100%)	T**G**A (100%)	T**G**A (97%)	T**G**A (100%)	AGU→S_2_ ACU→T_2_	n.a.	n.a.	Tolerated (0.83)
**11,461,801**	Exon 3 (PGA)	G**C**T	G**C**T (98%)	G**C**T (97%)	G**T**T (13%)	G**C**T (100%)	CGA→R_23_ CAA→Q_23_	n.a.	n.a.	Tolerated (0.57)
**11,461,772**	Exon 3 (PGA)	**G**CA	**G**CA (100%)	**G**CA (96%)	**A**CA (12%)	**G**CA (100%)	CGU→R_33_ UGU→C_33_	rs77775235	A = 0%	Tolerated (0.06)
**11,461,769**	Exon 3 (PGA)	**G**GG	**T**GG (5%) *	**T**GG (9%) *	**T**GG (5%) *	**T**GG (13%)	CCC→P_34_ ACC→T_34_	rs144658455	T = 0%	Tolerated (0.53)
**11,461,745**	Exon 3 (PGA)	**G**TT	**C**TT (8%) *	**C**TT (8%) *	**C**TT (5%) *	**C**TT (12%)	CAA→Q_42_ GAA→E_42_	rs76859544	C = 6.8%	Tolerated (1)
**11,461,742**	Exon 3 (PGA)	**C**CT	**T**CT (10%) *	**T**CT (27%)	**T**CT (11%)	**T**CT (7%) *	GGA→G_43_ AGA→R_43_	rs776943151	T = 0.05%	Tolerated (0.45)
**11,461,706**	Exon 3 (PGA)	**G**GG	**T**GG (14%)	**T**GG (23%)	**T**GG (13%)	**T**GG (20%)	CCC→P_55_ ACC→T_55_	rs12308381	T = 21.6%	Tolerated (0.12)
**11,461,675**	Exon 3 (PGA)	G**C**T	G**G**T (1%) *	G**G**T (2%) *	G**G**T (2%) *	G**G**T (28%)	CGA→R_65_ CCA→P_65_	rs75743553	G = 0%	Tolerated (0.32)
**11,461,673**	Exon 3 (PGA)	**G**GG	**G**GG (99%)	**A**GG (13%)	**A**GG (2%) *	**G**GG (100%)	CCC→P_66_ UCC→S_66_	rs1332850459	A = 0%	Tolerated (0.25)
**11,461,580**	Exon 3 (PGA)	**T**GG	**G**GG (65%)	**G**GG (52%)	**G**GG (24%)	**G**GG (54%)	ACC→T_97_ CCC→P_97_	n.a.	n.a.	Tolerated (0.81)
**11,461,570**	Exon 3 (PGA)	G**G**A	G**T**A (51%)	G**T**A (54%)	G**T**A (8%) *	G**T**A (47%)	CCU→P_100_ CAU→H_100_	n.a.	n.a.	Tolerated (0.59)
**11,461,553**	Exon 3 (PGA)	**T**CT	**C**CT (13%)	**C**CT (15%)	**T**CT (100%)	**C**CT (24%)	AGA→R_106_ GGA→G_106_	n.a.	n.a.	Tolerated (0.84)
**11,461,550**	Exon 3 (PGA)	**G**GT	**G**GT (100%)	**A**GT (17%)	**G**GT (100%)	**G**GT (100%)	CCA→P_107_ UCA→S_107_	n.a.	n.a.	Tolerated (0.50)
**11,461,549**	Exon 3 (PGA)	G**G**T	G**C**T (13%)	G**C**T (6%) *	G**G**T (100%)	G**C**T (13%)	CCA→P_107_ CGA→R_107_	n.a.	n.a.	Tolerated (0.9)
**11,461,525**	Exon 3 (PGA)	A**G**G	A**G**G (100%)	A**A**G (100%)	A**A**G (100%)	A**G**G (100%)	UCC→S_115_ UUC→F_115_	n.a.	n.a.	Damaging (0.04)
**11,461,513**	Exon 3 (PGA)	G**G**T	G**G**T (100%)	G**A**T (10%) *	G**A**T (11%)	G**G**T (100%)	CCA→_P119_ CUA→L_119_	n.a.	n.a.	Damaging (0.04)
**11,461,471**	Exon 3 (PGA)	C**C**A	C**C**A (100%)	C**T**A (4%) *	C**T**A (14%)	C**C**A (100%)	GGU→G_133_ GAU→D_133_	n.a.	n.a.	Tolerated (0.46)
**11,461,421**	Exon 3 (PGA)	**G**GG	**G**GG (100%)	**A**GG (5%) *	**A**GG (6%) *	**A**GG (100%)	CCC→P_150_ UCC→S_150_	n.a.	n.a.	Tolerated (0.18)
**11,461,420**	Exon 3 (PGA)	G**G**G	G**G**G (100%)	G**A**G (11%)	G**G**G (100%)	G**G**G (100%)	CCC→P_150_ CUC→L_150_	n.a.	n.a.	Tolerated (0.1)
**11,461,412**	Exon 3 (PGA)	**C**TT	**C**TT (100%)	**T**TT (14%)	**C**TT (100%)	**C**TT (100%)	GAA→E_153_ AAA→K_153_	n.a.	n.a.	Tolerated (0.85)
**11,461,319**	Exon 4 (P-D P32A)	**G**GA	**G**GA (97%)	**A**GA (9%) *	**A**GA (11%)	**G**GA (100%)	CCU→P_23_ UCU→S_23_	n.a.	n.a.	Tolerated (0.55)
**11,461,309**	Exon 4 (P-D P32A)	G**G**T	G**G**T (100%)	G**G**T (100%)	G**A**T (11%)	G**G**T (100%)	CCA→P_26_ CUA→L_26_	n.a.	n.a.	Damaging (0.01)
**11,461,229**	Exon 4 (P-D P32A)	**G**GA	**G**GA (100%)	**A**GA (13%)	**A**GA (4%) *	**G**GA (100%)	CCU→P_54_ UCU→S_54_	n.a.	n.a.	Tolerated (0.13)

^a^: Frequency of the substitution (highlighted bases) in the ancient hominin species, as reported in IGV considering the depth (coverage) of the reads displayed at the corresponding locus; * frequency ≤ 10% and ** counts < 10; n.a.: not available; NS: not scored. The variants fixed at 100% in modern humans compared with ancient hominines are highlighted in light orange. The genomic variants whose frequencies show a different geographic distribution among humans are in red text.

**Table 2 ijms-24-15010-t002:** Neanderthal and Denisovan nucleotide substitutions and the corresponding SIFT results on *PRH2*, *HTN1*, *HTN3*, *AMY1A*, *STATH*, and *SMR3B* gene loci.

Chromosome Position (hg19)	Gene Region	Modern Human	Altai Neanderthal (Variant Frequency ^a^)	Chagyrskaya Neanderthal (Variant Frequency ^a^)	Vindija Neanderthal (Variant Frequency ^a^)	Denisovan (Variant Frequency ^a^)	Codon→Amino Acid	SNP id	SNP Total Frequency (ALFA)	SIFT Results (Score)
***PRH2* (direct reading, chromosome 12)**										
**11,082,885**	Exon 2 (PRP-1)	**G**TT	**A**TT (2%) *	**A**TT (12%)	**A**TT (4%) *	**G**TT (100%)	GUU→V_12_ AUU→I_12_	rs776898585	A = 0%	N.S
**11,082,894**	Exon 2 (PRP-1)	**G**TA	**G**TA (100%)	**A**TA (12%)	**A**TA (10%) *	**G**TA (100%)	GUA→V_15_ AUA→I_15_	n.a.	n.a.	Tolerated (0.26)
**11,083,305**	Exon 3 (PRP-1)	**C**CA	**C**CA (98%)	**T**CA (14%)	**T**CA (14%)	**C**CA (100%)	CCA→P_33_ UCA→S_33_	n.a.	n.a.	Tolerated (0.07)
**11,083,318**	Exon 3 (PRP-1)	G**G**A	G**G**A (100%)	G**A**A (14%)	G**G**A (100%)	G**G**A (100%)	GGA→G_37_ GAA→E_37_	n.a.	n.a.	Tolerated (0.07)
**11,083,323**	Exon 3 (PRP-1)	**C**AA	**C**AA (100%)	**T**AA (8%) *	**T**AA (12%)	**C**AA (100%)	CAA→Q_39_ UAA→stop	n.a.	n.a.	Damaging due to stop
**11,083,426**	Exon 3 (PRP-1)	G**G**A	G**G**A (100%)	G**G**A (100%)	G**A**A (11%)	G**G**A (100%)	GGA→G_73_ GAA→E_73_	n.a.	n.a.	Damaging (0.02)
**11,083,431**	Exon 3 (PRP-1)	**C**CA	**C**CA (100%)	**T**CA (13%)	**T**CA (8%) *	**T**CA (6%) *	CCA→P_75_ UCA→S_75_	n.a.	n.a.	Tolerated (0.23)
**11,083,452**	Exon 3 (PRP-1)	**G**GA	**G**GA (100%)	**A**GA (6%) *	**A**GA (14%)	**G**GA (100%)	GGA→G_82_ AGA→R_82_	n.a.	n.a.	Damaging (0.01)
**11,083,455**	Exon 3 (PRP-1)	**G**GC	**G**GC (100%)	**A**GC (17%)	**G**GC (100%)	**G**GC (100%)	GGC→G_83_ AGC→S_83_	n.a.	n.a.	N.S.
**11,083,488**	Exon 3 (PRP-1)	**G**GA	**G**GA (100%)	**G**GA (100%)	**A**GA (11%)	**G**GA (100%)	GGA→G_94_ AGA→R_94_	n.a.	n.a.	Damaging (0.04)
**11,083,531**	Exon 3 (PRP-1)	A**G**G	A**G**G (100%)	A**G**G (100%)	A**A**G (18%)	A**G**G (100%)	AGG→R_108_ AAG→K_108_	n.a.	n.a.	N.S.
**11,083,536**	Exon 3 (PRP-1)	**C**AA	**C**AA (100%)	**T**AA (11%)	**C**AA (100%)	**C**AA (100%)	CAA→Q_110_ UAA→stop	n.a.	n.a.	N.S.
**11,083,545**	Exon 3 (PRP-1)	**C**CC	**C**CC (100%)	**T**CC (12%)	**T**CC (6%) *	**C**CC (100%)	CCC→P_113_ UCC→S_113_	rs1289206423	T = 0%	N.S.
**11,083,551**	Exon 3 (PRP-1)	**C**AG	**C**AG (97%)	**C**AG (100%)	**T**AG (13%)	**C**AG (100%)	CAG→Q_115_ UAG→stop	n.a.	n.a.	N.S.
**11,083,570**	Exon 3 (PRP-1)	G**G**T	G**G**T (100%)	G**A**T (18%)	G**G**T (100%)	G**G**T (100%)	GGU→G_121_ GAU→D_121_	n.a.	n.a.	N.S.
**11,083,575**	Exon 3 (PRP-1)	**C**CC	**C**CC (96%)	**T**CC (8%) *	**T**CC (15%)	**C**CC (100%)	CCC→P_123_ UCC→S_123_	n.a.	n.a.	N.S.
**11,083,581**	Exon 3 (PRP-1)	**C**CT	**C**CT (100%)	**T**CT (20%)	**T**CT (8%) *	**C**CT (100%)	CCU→P_125_ UCU→S_125_	n.a.	n.a.	N.S.
**11,083,582**	Exon 3 (PRP-1)	C**C**T	C**C**T (100%)	C**T**T (13%)	C**T**T (8%) *	C**C**T (100%)	CCU→P_125_ CUU→L_125_	n.a.	n.a.	N.S.
**11,083,605**	Exon 3 (PRP-1)	**C**CA	**C**CA (100%)	**T**CA (11%)	**C**CA (100%)	**C**CA (100%)	CCA→P_133_ UCA→S_133_	rs1343870622	T = 0%	N.S.
**11,083,618**	Exon 3 (PRP-1)	G**G**G	G**G**G (100%)	G**A**G (11%)	G**G**G (100%)	G**G**G (100%)	GGG→G_137_ GAG→E_137_	n.a.	n.a.	N.S.
**11,083,635**	Exon 3 (PRP-1)	**C**CT	**C**CT (100%)	**C**CT (100%)	**T**CT (16%)	**C**CT (100%)	CCU→P_143_ UCU→S_143_	n.a.	n.a.	N.S.
**11,083,636**	Exon 3 (PRP-1)	C**C**T	C**C**T (100%)	C**C**T (100%)	C**T**T (11%)	C**C**T (100%)	CCU→P_143_ CUU→L_143_	n.a.	n.a.	N.S.
**11,083,663**	Exon 3 (C-term removal)	T**C**T	T**C**T (100%)	T**C**T (100%)	T**T**T (17%)	T**C**T (100%)	UCU→S_152(rem)_ UUU→F_152(rem)_	rs746351335	n.a.	N.S.
***HTN1* (direct reading, chromosome 4)**										
**70,920,165**	Exon 4	**C**AT	**C**AT (100%)	**T**AT (2%) *	**T**AT (13%)	**C**AT (100%)	CAU**→**H_15_ UAU**→**Y_15_	n.a.	n.a.	Tolerated (0.37)
**70,921,215**	Exon 5	**G**AA	**G**AA (100%)	**A**AA (3%) *	**A**AA (11%)	**G**AA (100%)	GAA**→**E_16_ AAA**→**K_16_	n.a.	n.a.	N.S
**70,921,234**	Exon 5	C**G**A	C**A**A (2%) *	C**A**A (58%)	C**A**A (3%) *	C**G**A (100%)	CGA**→**R_32_ CAA**→**Q_32_	rs375127098	A = 0.014%	N.S
***HTN3* (direct reading, chromosome 4)**										
**70,896,460**	Exon 2 (Signal)	AT**G**	AT**G** (100%)	AT**A** (11%)	AT**G** (100%)	AT**G** (100%)	AUG**→**M_0(sp)_ AUA**→**I_0(sp)_	n.a.	n.a.	N.S
**70,897,696**	Exon 3 (Signal)	**G**GA	**G**GA (100%)	**A**GA (12%)	**A**GA (4%) *	**G**GA (100%)	GGA**→**G_17(sp)_ AGA**→**R_17(sp)_	rs1254624179	n.a.	N.S
***AMY1A* (reverse reading, chromosome 1)**										
**104,238,248**	Exon 2 (Signal)	A**C**C	A**C**C (100%)	A**C**C (100%)	A**T**C (15%)	A**C**C (100%)	UGG→W_4(sp)_ UAG→stop	n.a.	n.a.	Damaging due to stop
**104,238,189**	Exon 2	**G**CT	**G**CT (100%)	**A**CT (13%)	**A**CT (20%) **	**G**CT (100%)	CGA→R_10_ UGA→stop	n.a.	n.a.	Damaging due to stop
**104,237,696**	Exon 3	A**C**C	A**C**C (100%)	A**C**C (100%)	A**T**C (17%)	A**C**C (100%)	UGG→W_59_ UAG→stop	n.a.	n.a.	Damaging due to stop
**104,237,685**	Exon 3	**G**TT	**G**TT (100%)	**G**TT (100%)	**A**TT (14%)	**G**TT (100%)	CAA→Q_63_ UAA→stop	n.a.	n.a.	Damaging due to stop
**104,237,626**	Exon 3	TA**C**	TA**C** (100%)	TA**C** (100%)	TA**T** (15%)	TA**C** (100%)	AUG→M_82_ AUA→I_82_	n.a.	n.a.	Damaging (0.01)
**104,236,795**	Exon 4	**G**CA	**G**CA (100%)	**G**CA (100%)	**A**CA (13%)	**G**CA (100%)	CGU→R_92_ UGU→C_92_	n.a.	n.a.	Damaging (0)
**104,236,666**	Exon 4	**C**TA	**C**TA (100%)	**C**TA (100%)	**T**TA (11%)	**C**TA (100%)	GAU→D_135_ AAU→N_135_	n.a.	n.a.	Tolerated (0.08)
**104,236,654**	Exon 4	**C**CA	**C**CA (100%)	**T**CA (5%) *	**T**CA (11%)	**C**CA (100%)	GGU→G_139_ AGU→S_139_	n.a.	n.a.	Tolerated (0.6)
**104,236,152**	Exon 5	**C**AG	**C**AG (100%)	**T**AG (15%)	**T**AG (20%)	**C**AG (100%)	GUC→V_157_ AUC→I_157_	n.a.	n.a.	Tolerated (0.17)
**104,236,146**	Exon 5	**C**TA	**C**TA (100%)	**T**TA (8%) *	**T**TA (12%)	**C**TA (100%)	GAU→D_159_ AAU→N_159_	n.a.	n.a.	Tolerated (1)
**104,236,139**	Exon 5	G**C**A	G**T**A (4%) *	G**T**A (7%) *	G**T**A (12%)	G**C**A (100%)	CGU→R_161_ CAU→H_161_	n.a.	n.a.	Damaging (0.01)
**104,236,080**	Exon 5	**C**TT	**C**TT (100%)	**C**TT (100%)	**T**TT (13%)	**C**TT (100%)	GAA→E_181_ AAA→K_181_	n.a.	n.a.	Tolerated (0.11)
**104,235,996**	Exon 5	**C**GT	**C**GT (96%)	**C**GT (100%)	**T**GT (13%)	**C**GT (100%)	GCA→A_209_ ACA→T_209_	n.a.	n.a.	Tolerated (0.27)
**104,235,164**	Exon 6	**C**TC	**C**TC (100%)	**C**TC (100%)	**T**TC (11%)	**C**TC (100%)	GAG→E_240_ AAG→K_240_	n.a.	n.a.	Damaging (0.01)
**104,235,148**	Exon 6	T**C**A	T**C**A (100%)	T**C**A (100%)	T**T**A (18%)	T**C**A (100%)	AGU→S_245_ AAU→N_245_	n.a.	n.a.	Tolerated (0.52)
**104,235,083**	Exon 6	**G**CG	**A**CG (3%) *	**A**CG (6%) *	**A**CG (12%)	**G**CG (100%)	CGC→R_267_ UGC→C_267_	n.a.	n.a.	Damaging (0)
**104,234,224**	Exon 7	C**C**T	C**C**T (100%)	C**C**T (100%)	C**T**T (13%)	C**C**T (100%)	GGA→G_281_ GAA→E_281_	n.a.	n.a.	Damaging (0)
**104,234,218**	Exon 7	C**C**A	C**C**A (100%)	C**T**A (13%)	C**T**A (15%)	C**C**A (100%)	GGU→G_283_ GAU→D_283_	n.a.	n.a.	Tolerated (0.25)
**104,234,129**	Exon 7	**G**AA	**G**AA (100%)	**A**AA (13%)	**G**AA (100%)	**G**AA (100%)	CUU→L_313_ UUU→F_313_	n.a.	n.a.	Damaging (0)
**104,234,125**	Exon 7	T**G**G	T**G**G (100%)	T**A**G (17%)	T**G**G (100%)	T**G**G (100%)	ACC→T_314_ AUC→I_314_	n.a.	n.a.	Damaging (0)
**104,233,978**	Exon 8	**G**GA	**G**GA (100%)	**A**GA (13%)	**A**GA (11%)	**G**GA (100%)	CCU→P_332_ UCU→S_332_	n.a.	n.a.	Damaging (0.05)
**104,233,977**	Exon 8	G**G**A	G**G**A (100%)	G**A**A (6%) *	G**A**A (11%)	G**G**A (100%)	CCU→P_332_ CUU→L_332_	n.a.	n.a.	Damaging (0)
**104,233,963**	Exon 8	**G**CT	**G**CT (100%)	**G**CT (100%)	**A**CT (14%)	**G**CT (100%)	CGA→R_337_ UGA→stop	rs19955486	A = 0.08%	Damaging due to stop
**104,231,858**	Exon 9	A**C**A	A**C**A (100%)	A**C**A (100%)	A**T**A (11%)	A**C**A (100%)	UGU→C_378_ UAU→Y_378_	n.a.	n.a.	Damaging (0)
**104,231,680**	Exon 10	**C**AC	**C**AC (100%)	**T**AC (4%) *	**T**AC (20%)	**C**AC (100%)	GUG→V_401_ AUG→M_401_	n.a.	n.a.	Damaging (0)
**104,231,643**	Exon 10	C**C**C	C**C**C (100%)	C**T**C (5%) *	C**T**C (11%)	C**C**C (100%)	GGG→G_413_ GAG→E_413_	n.a.	n.a.	Damaging (0.02)
**104,231,622**	Exon 10	C**C**C	C**C**C (100%)	C**C**C (100%)	C**T**C (13%)	C**C**C (100%)	GGG→G_420_ GAG→E_420_	n.a.	n.a.	Tolerated (0.08)
**104,230,237**	Exon 11	T**G**A	T**G**A (100%)	T**G**A (100%)	T**A**A (13%)	T**G**A (100%)	ACU→T_442_ AUU→I_442_	n.a.	n.a.	Damaging (0)
**104,230,129**	Exon 11	A**G**A	A**G**A (100%)	A**G**A (100%)	A**A**A (13%)	A**G**A (100%)	UCU→S_478_ UUU→F_478_	n.a.	n.a.	Tolerated (0.62)
***STATH* (direct reading, chromosome 4)**										
**70,866,583**	Exon 5	**G**GG	**G**GG (100%)	**A**GG (13%)	**A**GG (3%) *	**G**GG (100%)	GGG**→**G_17_ AGG**→**R_17_	n.a.	n.a.	N.A.
**70,866,616**	Exon 5	**C**CA	**C**CA (98%)	**C**CA (100%)	**T**CA (11%)	**T**CA (3%) *	CCA**→**P_28_ UCA**→**S_28_	n.a.	n.a.	N.A.
**70,866,626**	Exon 5	C**C**A	C**C**A (100%)	C**T**A (15%)	C**C**A (100%)	C**C**A (96%)	CCA**→**P_31_ CUA**→**L_31_	n.a.	n.a.	N.A.
**70,866,628**	Exon 5	**C**AA	**C**AA (100%)	**T**AA (15%)	**C**AA (100%)	**C**AA (100%)	CAA**→**Q_32_ UAA**→**stop	n.a.	n.a.	Damaging due to stop
***SMR3B* (direct reading, chromosome 4)**										
**71,255,405**	Exon 3	A**G**G	A**G**G (100%)	A**G**G (100%)	A**A**G (12%)	A**G**G (100%)	AGG→R_5_ AAG→K_5_	rs777831757	A = 0%	NS
**71,255,444**	Exon 3	C**C**T	C**C**T (100%)	C**T**T (12%)	C**T**T (3%) *	C**C**T (100%)	CCU→P_18_ CUU→L_18_	n.a.	n.a.	NS
**71,255,495**	Exon 3	G**G**G	G**G**G (100%)	G**G**G (94%)	G**A**G (17%)	G**G**G (100%)	GGG→G_35_ GAG→E_35_	n.a.	n.a.	NS

^a^: Frequency of the substitution (highlighted bases) in the ancient hominin species, as reported in IGV considering the depth (coverage) of the reads displayed at the corresponding locus; * frequency ≤ 10% and ** counts < 10; n.a.: not available; NS: not scored.

**Table 3 ijms-24-15010-t003:** Neanderthal and Denisovan nucleotide substitutions and the corresponding SIFT results on *CST1*, *CST2*, *CST3*, *CST4*, *CST5*, *CSTA*, and *CSTB* gene loci.

Chromosome Position (hg19)	Gene Region	Modern Human	Altai Neanderthal (Variant Frequency ^a^)	Chagyrskaya Neanderthal (Variant Frequency ^a^)	Vindija Neanderthal (Variant Frequency ^a^)	Denisovan (Variant Frequency ^a^)	Codon→Amino Acid	SNP id	SNP Total Frequency (ALFA)	SIFT Results (Score)
***CST1* (reverse reading, chromosome 20)**										
**23,731,494**	Exon 1 (Signal)	**A**TA	**G**TA (100%)	**G**TA (95%)	**G**TA (100%)	**G**TA (100%)	UAU→Y_3(sp)_ CAU→H_3(sp)_	rs6076122	G = 71.1%	Tolerated (0.11)
**23,731,463**	Exon 1 (Signal)	T**G**G	T**A**G (2%) *	T**A**G (13%)	T**A**G (5%) *	T**G**G (100%)	ACC→T_13(sp)_ AUC→I_13(sp)_	n.a.	n.a.	Tolerated (0.39)
**23,731,455**	Exon 1 (Signal)	**C**AC	**C**AC (100%)	**C**AC (100%)	**T**AC (16%)	**C**AC (100%)	GUG→V_16(sp)_ AUG→M_16(sp)_	n.a.	n.a.	Tolerated (0.23)
**23,731,446**	Exon 1 (Signal)	**C**GG	**C**GG (100%)	**C**GG (100%)	**T**GG (11%)	**C**GG (100%)	GCC→A_19(sp)_ ACC→T_19(sp)_	rs1425228752	T = 0.001%	Damaging (0.01)
**23,731,439**	Exon 1	T**C**G	T**C**G (100%)	T**T**G (6%) *	T**T**G (14%)	T**C**G (100%)	AGC→S_2_ AAC→N_2_	n.a.	n.a.	Tolerated (0.15)
**23,731,428**	Exon 1	**C**TC	**C**TC (100%)	**C**TC (100%)	**T**TC (21%)	**C**TC (100%)	GAG→E_6_ AAG→K_6_	rs1292698911	T = 0.0004%	Tolerated (0.66)
**23,731,394**	Exon 1	C**G**T	C**G**T (100%)	C**A**T (13%)	C**G**T (100%)	C**G**T (100%)	GCA→A_17_ GUA→V_17_	n.a.	n.a.	Tolerated (0.25)
**23,731,344**	Exon 1	**C**TC	**T**TC (3%) *	**C**TC (100%)	**T**TC (11%)	**T**TC (3%) *	GAG→E_34_ AAG→K_34_	rs368203290	T = 0.008%	Tolerated (0.07)
**23,731,307**	Exon 1	G**C**A	G**C**A (100%)	G**T**A (14%)	G**C**A (100%)	G**T**A (6%) *	CGU→R_46_ CAU→H_46_	rs758187154	T = 0%	Damaging (0.01)
**23,731,281**	Exon 1	**G**TT	**G**TT (100%)	**G**TT (100%)	**A**TT (13%)	**G**TT (100%)	CAA→Q_55_ UAA→stop	n.a.	n.a.	Damaging due to stop
**23,729,759**	Exon 2	C**C**C	C**C**C (100%)	C**C**C (100%)	C**G**C (26%)	C**C**C (100%)	GGG→G_59_ GCG→A_59_	n.a.	n.a.	Tolerated (1)
**23,729,700**	Exon 2	**G**GG	**G**GG (100%)	**G**GG (100%)	**A**GG (11%)	**G**GG (100%)	CCC→P_79_ UCC→S_79_	n.a.	n.a.	Tolerated (0.38)
**23,729,699**	Exon 2	G**G**G	G**G**G (100%)	G**A**G (3%) *	G**A**G (11%)	G**G**G (100%)	CCC→P_79_ CUC→L_79_	rs756782667	A = 0%	Tolerated (0.06)
**23,729,687**	Exon 2	T**G**G	T**G**G (100%)	T**A**G (16%)	T**A**G (4%) *	T**G**G (100%)	ACC→T_83_ AUC→I_83_	n.a.	n.a.	Damaging (0.02)
**23,728,503**	Exon 3	**G**GG	**G**GG (100%)	**A**GG (11%)	**A**GG (3%) *	**G**GG (100%)	CCC→P_106_ UCC→S_106_	rs754531104	A = 0.004%	Tolerated (0.09)
**23,728,494**	Exon 3 (Cys-SN)	**T**TG	**C**TG (10%) *	**C**TG (11%)	**C**TG (14%)	**C**TG (4%) *	AAC→N_109_ GAC→D_109_	rs3188319	C = 0.004%	Tolerated (1)
**23,728,490**	Exon 3	T**C**T	T**T**T (2%) *	T**T**T (14%)	T**C**T (100%)	T**C**T (100%)	AGA→R_110_ AAA→K_110_	n.a.	n.a.	Tolerated (1)
**23,728,487**	Exon 3	T**C**C	T**C**C (100%)	T**T**C (13%)	T**T**C (7%) *	T**C**C (100%)	AGG→R_111_ AAG→K_111_	rs3188320	T = 0%	Tolerated (0.85)
***CST2* (reverse reading, chromosome 20)**										
**23,807,260**	Exon 1 (Signal)	C**G**G	C**G**G (100%)	C**G**G (100%)	C**A**G (14%)	C**G**G (100%)	GCC→A_12(sp)_ GUC→V_12(sp)_	rs1411653443	A = 0.007%	Damaging (0.02)
**23,807,257**	Exon 1 (Signal)	T**G**G	T**G**G (100%)	T**A**G (14%)	T**G**G (100%)	T**G**G (100%)	ACC→T_13(sp)_ AUC→I_13(sp)_	n.a.	n.a.	Tolerated (0.43)
**23,807,245**	Exon 1 (Signal)	C**G**G	C**G**G (100%)	C**A**G (14%)	C**G**G (100%)	C**G**G (100%)	GCC→A_17(sp)_ GUC→V_17(sp)_	n.a.	n.a.	Tolerated (0.1)
**23,807,231**	Exon 1	**G**GG	**G**GG (100%)	**A**GG (14%)	**A**GG (8%) *	**G**GG (100%)	CCC→P_3_ UCC→S_3_	n.a.	n.a.	Tolerated (1)
**23,807,162**	Exon 1	**G**CA	**A**CA (95%)	**A**CA (100%)	**A**CA (100%)	**A**CA (8%) *	CGU→R_26_ UGU→C_26_	rs111349461	A = 0.06%	Damaging (0.05)
**23,807,138**	Exon 1	**C**TC	**T**TC (3%) *	**T**TC (12%)	**T**TC (6%) *	**C**TC (100%)	GAG→E_34_ AAG→K_34_	rs541427772	T = 0.017%	Tolerated (0.07)
**23,807,102**	Exon 1	**G**CG	**A**CG (3%) *	**G**CG (100%)	**A**CG (11%)	**G**CG (100%)	CGC→R_46_ UGC→C_46_	rs112783512	A = 0.019%	Tolerated (0.07)
**23,807,093**	Exon 1	**G**CC	**G**CC (100%)	**A**CC (4%)	**A**CC (20%)	**G**CC (100%)	CGG→R_49_ UGG→W_49_	rs55860552	A = 0.12%	Damaging (0)
**23,807,084**	Exon 1	**G**CT	**G**CT (100%)	**A**CT (5%) *	**A**CT (15%)	**G**CT (100%)	CGA→R_52_ UGA→stop	rs568411970	A = 0%	Damaging due to stop
**23,807,077**	Exon 1	T**C**C	T**C**C (100%)	T**C**C (100%)	T**T**C (13%)	T**C**C (100%)	AGG→R_54_ AAG→K_54_	n.a.	n.a.	Tolerated (0.34)
**23,807,075**	Exon 1	**C**TC	**C**TC (100%)	**T**TC (12%)	**T**TC (12%)	**C**TC (100%)	GAG→E_55_ AAG→K_55_	n.a.	n.a.	Tolerated (1)
**23,805,930**	Exon 2	**T**AT	**C**AT (7%) *	**C**AT (5%) *	**C**AT (14%)	**C**AT (4%) *	AUA→I_67_ GUA→V_67_	rs199856966	C = 0.004%	Tolerated (1)
**23,805,917**	Exon 2	G**C**T	G**T**T (2%) *	G**T**T (13%)	G**T**T (5%) *	G**T**T (2%) *	CGA→R_71_ CAA→Q_71_	rs150428155	T = 0.008%	Damaging (0.01)
**23,805,878**	Exon 2	A**C**A	A**C**A (100%)	A**C**A (97%)	A**T**A (14%)	A**C**A (100%)	UGU→C_84_ UAU→Y_84_	n.a.	n.a.	Damaging (0)
**23,805,875**	Exon 2	C**G**G	C**G**G (100%)	C**A**G (15%)	C**A**G (2%) *	C**G**G (100%)	GCC→A_85_ GUC→V_85_	n.a.	n.a.	Tolerated (0.06)
**23,804,730**	Exon 3	A**C**G	A**C**G (100%)	A**T**G (7%) *	A**T**G (11%)	A**C**G (100%)	UGC→C_98_ UAC→Y_98_	n.a.	n.a.	Damaging (0)
**23,804,702**	Exon 3	AC**C**	AC**C** (100%)	AC**T** (12%)	AC**C** (100%)	AC**C** (100%)	UGG→W_107_ UGA→stop	rs1380420803	n.a.	Damaging due to stop
**23,804,691**	Exon 3	T**A**C	T**C**C (13%)	T**C**C (10%) *	T**C**C (9%) *	T**A**C (100%)	AUG→M_111_ AGG→R_111_	rs202150666	C = 0.01%	Tolerated (0.31)
***CST3* (reverse reading, chromosome 20)**										
**23,618,472**	Exon 1 (Signal)	**G**AG	**G**AG (100%)	**A**AG (8%) *	**A**AG (15%)	**G**AG (100%)	CUC→L_8(sp)_ UUC→F_8(sp)_	rs1285248919	n.a.	Damaging (0)
**23,618,433**	Exon 1	**G**GG	**G**GG (100%)	**G**GG (100%)	**A**GG (13%)	**G**GG (100%) **	CCC→P_22(sp)_ UCC→S_22(sp)_	n.a.	n.a.	Tolerated (0.5)
**23,618,370**	Exon 1	**C**AC	**C**AC (100%)	**C**AC (100%)	**T**AC (13%)	**C**AC (100%)	GUG→V_18_ AUG→M_18_	n.a.	n.a.	Tolerated (0.11)
**23,618,358**	Exon 1	**C**CA	**C**CA (100%)	**T**CA (22%)	**T**CA (4%) *	**C**CA (100%)	GGU→G_22_ AGU→S_22_	n.a.	n.a.	Tolerated (0.48)
**23,618,357**	Exon 1	C**C**A	C**C**A (100%)	C**T**A (11%)	C**C**A (100%)	C**C**A (100%)	GGU→G_22_ GAU→D_22_	n.a.	n.a.	Tolerated (0.56)
**23,618,295**	Exon 1	**G**TG	**G**TG (100%)	**G**TG (100%)	**A**TG (13%)	**G**TG (100%)	CAC→H_43_ UAC→Y_43_	n.a.	n.a.	Tolerated (1)
**23,615,994**	Exon 2	C**C**C	C**T**C (3%) *	C**C**C (100%)	C**T**C (13%)	C**C**C (100%)	GGG→G_59_ GAG→E_59_	n.a.	n.a.	Damaging (0.01)
**23,614,564**	Exon 3	**G**TC	**G**TC (100%)	**G**TC (100%)	**A**TC (13%)	**G**TC (100%)	CAG→Q_118_ UAG→stop	n.a.	n.a.	Damaging due to stop
***CST4* (reverse reading, chromosome 20)**										
**23,669,566**	Exon 1 (Signal)	T**G**G	T**G**G (100%)	T**A**G (7%) *	T**A**G (11%)	T**G**G (100%)	ACC→T_13(sp)_ AUC→I_13(sp)_	rs770415022	n.a.	Tolerated (0.37)
**23,669,561**	Exon 1 (Signal)	**C**GA	**C**GA (100%)	**C**GA (100%)	**C**GA (100%)	**A**GA (100%)	GCU→A_15(sp)_ UCU→S_15(sp)_	n.a.	n.a.	Tolerated (0.39)
**23,669,539**	Exon 1	A**G**G	A**G**G (100%)	A**A**G (5%) *	A**A**G (13%)	A**G**G (100%)	UCC→S_3_ UUC→F_3_	n.a.	n.a.	Tolerated (0.08)
**23,669,470**	Exon 1	G**C**A	G**C**A (100%)	G**T**A (15%)	G**C**A (100%)	G**T**A (17%)	CGU→R_26_ CAU→H_26_	rs201273557	T = 0.01%	Tolerated (0.08)
**23,669,462**	Exon 1	**G**TG	**G**TG (100%)	**G**TG (100%)	**A**TG (18%)	**G**TG (100%)	CAC→H_29_ UAC→Y_29_	n.a.	n.a.	Tolerated (0.06)
**23,669,408**	Exon 1	**G**GC	**G**GC (100%)	**A**GC (12%)	**G**GC (100%)	**G**GC (100%)	CCG→P_47_ UCG→S_47_	n.a.	n.a.	Tolerated (0.06)
**23,667,835**	Exon 2	**A**AA	**C**AA (97%)	**C**AA (100%)	**C**AA (90%)	**A**AA (100%)	UUU→F_58_ GUU→V_58_	rs145608577	C = 0.2%	Tolerated (1)
**23,667,828**	Exon 2	C**C**C	C**C**C (100%)	C**T**C (18%)	C**C**C (100%)	C**C**C (100%)	GGG→G_60_ GAG→E_60_	rs144556333	T = 0.007%	Damaging (0)
**23,667,826**	Exon 2	**C**AC	**C**AC (100%)	**T**AC (10%) *	**T**AC (27%)	**C**AC (100%)	GUG→V_61_ AUG→M_61_	n.a.	n.a.	Tolerated (0.24)
**23,667,808**	Exon 2	**C**AT	**C**AT (100%)	**T**AT (13%)	**C**AT (100%)	**T**AT (4%) *	GUA→V_67_ AUA→I_67_	rs774067751	T = 0.007%	Tolerated (0.23)
**23,667,792**	Exon 2	T**G**G	T**G**G (100%)	T**A**G (13%)	T**G**G (100%)	T**G**G (100%)	ACC→T_72_ AUC→I_72_	n.a.	n.a.	Damaging (0)
**23,667,783**	Exon 2	T**G**G	T**G**G (100%)	T**G**G (95%)	T**A**G (15%)	T**G**G (100%)	ACC→T_75_ AUC→I_75_	rs760057501	A = 0%	Damaging (0.01)
**23,666,565**	Exon 3	T**A**C	T**C**C (88%)	T**C**C (14%)	T**C**C (80%)	T**A**C (100%)	AUG→M_111_ AGG→R_111_	rs779547810	C = 0%	Tolerated (0.87)
***CST5* (reverse reading, chromosome 20)**										
**23,860,243**	Exon 1	A**G**C	A**A**C (3%) *	A**G**C (100%)	A**A**C (11%)	A**A**C (5%) *	UCG→S_4_ UUG→L_4_	rs145031249	A = 0.011%	Tolerated (0.27)
**23,860,211**	Exon 1	**G**TA	**G**TA (100%)	**G**TA (100%)	**A**TA (12%)	**G**TA (100%)	CAU→H_15_ UAU→Y_15_	n.a.	n.a.	Tolerated (1)
**23,860,199**	Exon 1	**G**AG	**G**AG (100%)	**A**AG (11%)	**G**AG (100%)	**G**AG (100%)	CUC→L_19_ UUC→F_19_	rs370924959	A = 0%	Tolerated (0.66)
**23,860,178**	Exon 1	**A**CA	**G**CA (93%)	**G**CA (100%)	**G**CA (95%)	**G**CA (100%)	UGU→ C_26_ CGU→ R_26_	rs1799841	G = 43.2%	Tolerated (1)
**23,860,174**	Exon 1	C**G**G	C**G**G (100%)	C**G**G (100%)	C**A**G (11%)	C**G**G (100%)	GCC→A_27_ GUC→V_27_	n.a.	n.a.	Tolerated (0.18)
**23,860,130**	Exon 1	**C**TA	**C**TA (100%)	**C**TA (100%)	**T**TA (14%)	**C**TA (100%)	GAU→D_42_ AAU→N_42_	rs1257216384	n.a.	Tolerated (0.11)
**23,860,093**	Exon 1	C**G**G	C**G**G (100%)	C**G**G (100%)	C**A**G (11%)	C**G**G (100%)	GCC→A_54_ GUC→V_54_	n.a.	n.a.	Tolerated (0.11)
**23,858,200**	Exon 2	T**G**G	T**G**G (100%)	T**A**G (22%)	T**G**G (100%)	T**G**G (100%)	ACC→T_76_ AUC→I_76_	rs41282292	A = 0.061%	Damaging (0)
***CSTA* (direct reading, chromosome 3)**										
**122,044,197**	Exon 1	**G**TT	**G**TT (100%)	**A**TT (11%)	**G**TT (100%)	**G**TT (100%)	GUU→V_20_ AUU→I_20_	rs778366890	A = 0%	Tolerated (0.23)
**122,056,400**	Exon 2	**C**CA	**C**CA (100%)	**C**CA (100%)	**T**CA (12%)	**C**CA (100%)	CCA→P_25_ UCA→S_25_	n.a.	n.a.	Tolerated (0.74)
**122,060,361**	Exon 3	**C**TT	**C**TT (100%)	**C**TT (100%)	**T**TT (16%)	**C**TT (100%)	CUU→L_82_ UUU→F_82_	n.a.	n.a.	Damaging (0)
**122,060,373**	Exon 3	**C**AG	**C**AG (100%)	**C**AG (100%)	**T**AG (12%)	**C**AG (100%)	CAG→Q_86_ UAG→stop	n.a.	n.a.	Damaging due to stop
***CSTB* (reverse reading, chromosome 21)**										
**45,194,562**	Exon 2	**C**GC	**T**GC (2%) *	**T**GC (11%)	**C**GC (100%)	**C**GC (100%)	GCG→A_49_ ACG→T_49_	rs559906825	T = 0.007%	Damaging (0)
**45,194,138**	Exon 3	T**G**G	T**G**G (98%)	T**C**G (13%)	T**G**G (95%)	T**G**G (100%)	ACC→T_81_ AGC→S_81_	n.a.	n.a.	Tolerated (0.65)
**45,194,132**	Exon 3	A**G**A	A**G**A (100%)	A**G**A (100%)	A**A**A (15%)	A**G**A (100%)	UCU→S_83_ UUU→F_83_	n.a.	n.a.	Tolerated (0.1)

^a^: Frequency of the substitution (highlighted bases) in the ancient hominin species, as reported in IGV considering the depth (coverage) of the reads displayed at the corresponding locus; * frequency ≤ 10% and ** counts < 10; n.a.: not available. The variants fixed at 100% in modern humans compared with ancient hominines are highlighted in light orange. The genomic variants whose frequencies show a different geographic distribution among humans are in red text.

## Data Availability

All data reported in this manuscript are shown in the results section and further supported by the extended datasets provided in the Appendix A. No new primary datasets to be deposited have been generated.

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
