# Peer review of "A Catalog of Coding Sequence Variations in Salivary Proteins’ Genes Occurring during Recent Human Evolution"

_ijms, 2023, doi:10.3390/ijms241915010_

Round 1

Reviewer 1 Report (Previous Reviewer 1)

The manuscript has been significantly improved by extending the research to Neanderthals from different areas, as well as few additional salivary proteins, providing more reliable and comprehensive results. The results are well presented and explained. The only minor problem is the size of the Table 1 but I do not advise reducing it because it is difficult to choose which data to remove without compromising the integrity of the results. The Material and methods are described in sufficient detail.  Thera are only couple of minor issues that should be corrected.

I recommend adding short description/explanation of included Neanderthals, either in the introduction or in the discussion.

Line 176 add „that“ before „may“ for better clarity.

Line 284 sentence „Four of these…“ is unclear and should be rewritten.

Author Response

AUTHORS' REPLIES ARE IN BOLD

The manuscript has been significantly improved by extending the research to Neanderthals from different areas, as well as few additional salivary proteins, providing more reliable and comprehensive results. The results are well presented and explained. The only minor problem is the size of the Table 1 but I do not advise reducing it because it is difficult to choose which data to remove without compromising the integrity of the results. The Material and methods are described in sufficient detail.  Thera are only couple of minor issues that should be corrected.

I recommend adding short description/explanation of included Neanderthals, either in the introduction or in the discussion.

We thank the Reviewer for appreciating the revised version of the manuscript and for the additional tips provided. As suggested, we explained more about why we included in our analysis Neanderthals from different areas. Please refer to lines 528-534.

Line 176 add „that“ before „may“ for better clarity.

We have rephrased the sentence.

Line 284 sentence „Four of these…“ is unclear and should be rewritten.

We have rephrased the sentence.

Reviewer 2 Report (New Reviewer)

A somewhat straightforward study but a fairly successful one. I just have a few comments. 

I recommend removing the “whats in a neanderthals and denisovans kiss?’ from the title of the article. It was awkward and unnecessary

Figure 1 looks a bit odd, can you add size bars for some perspective?

What exactly was the protocol for picking THESE genes? I can’t seem to find it in the article.

Line 200 spelling

I think all the tables could be supplemental. It seems too big for the main article. Is there anyway to summarize these in some way? Perhaps average out the frequency /SIFT? Not quite sure how that would best be done.

In table 1 what exactly does the (variant frequency) refer to? Is that % within humans?

In the abstract: the final sentence says you found no sign of evo pressures or introgression. This is a bold statement that it doesn’t seem like you did any testing for. Also on 489-491. I suggest these assertions be removed from the text.

Didn't notice any issues aside from a spelling error.

Author Response

AUTHORS' REPLIES ARE IN BOLD

I recommend removing the “whats in a neanderthals and denisovans kiss?’ from the title of the article. It was awkward and unnecessary.

We thank the Reviewer for the suggestion. Accordingly, we propose a revised title for the manuscript as follow: “Identification of Coding Sequence Variations in Salivary Proteins’ Genes Occurred During Recent Human Evolution”.

Figure 1 looks a bit odd, can you add size bars for some perspective?

We thank the Reviewer for underlining this aspect and we have accordingly revised Figures 1 and 2 to improve clarity. We believe that size bars are not appropriate nor explicative for this kind of graphical representation of proteins, which are extremely polymorphic, with each different isoform having a different size and composition, as discussed in the introduction. Instead, we have revised and improved the clarity of legends, explaining the content of the figures.

What exactly was the protocol for picking THESE genes? I can’t seem to find it in the article.

We apologize for the lack of clarity, and tried to improve the introduction and discussion sections explaining the reason why we have focused on these protein families. The analyzed salivary proteins represent the most abundant and best characterized protein species in saliva, and have been also implicated as biomarkers for clinical conditions, suggesting their significant contribution to oral homeostasis and salivary functions (please refer to lines 61-65 and 114-117 in the Introduction section and to lines 435-437 in the Discussion section of the revised manuscript).

Line 200 spelling

The full name of the acronym SIFT was already spelled on lines 156-157 where it appears for the first time.

I think all the tables could be supplemental. It seems too big for the main article. Is there anyway to summarize these in some way? Perhaps average out the frequency /SIFT? Not quite sure how that would best be done.

We thank the Referee for this suggestion. We have already attempted to simplify as much as possible the complexity of data shown in the tables. Yet, we believe that given the type of study, the data shown in tables are key to understand our main results. In addition, we believe that the SIFT score is a functionally significant information (evaluation of the potential impact of nonsynonymous variants on salivary proteins’ function), which needs to be included. Based on these considerations, we believed not to remove the tables for the main text.

In table 1 what exactly does the (variant frequency) refer to? Is that % within humans?

We thank the Reviewer for raising this question and we have better described this point in all the tables’ captions and in the Materials and Methods section (lines 541-543). The variant frequency shown in table indicates the frequency (%) of that substitution in the ancient hominin species shown in the column header, as reported in the IGV tool, considering the depth (coverage) of the reads displayed at each locus.

In the abstract: the final sentence says you found no sign of evo pressures or introgression. This is a bold statement that it doesn’t seem like you did any testing for. Also on 489-491. I suggest these assertions be removed from the text.

These selective pressure analyses (as the population branch statistics analysis) were carried out during the first round of revision, and the corresponding results were described in paragraph “2.8. Evolutionary pressure of salivary protein genes” in the Results section and in paragraph “5.3. Selective pressure analysis” in the Materials and Methods section. Therefore, we do not agree to remove this sentence.

Reviewer 3 Report (New Reviewer)

The manuscript "What’s in the Neanderthal’s and Denisovan’s kisses? Archaic
coding sequence variants in ancient hominin salivary proteins" focuses on a set of genetic variants identified comparing archaic and modern humans, which are located in genes belonging to families known to be related to salivary function. I really like the idea of the research, since these genes might reveal interesting adaptations related to diet. However, the paper suffers from several limitations that in my opinion make it of insufficient quality for publication.

A first point is that while the different genes are discussed lengthily in the paper, there does not seem to be an objective criterion for their inclusion. Are they all the differentially expressed genes in saliva, are they based on some GO? It feels arbitrary, despite the long introduction. The lack of objective criteria also makes comparisons harder, particularly for statistical tests; though here this is paradoxically not a problem: there are little to no statistical tests. And here my major issue comes into play. The paper is pretty much devoid of evolutionary or genetic analyses. It is just a catalog out of publicly available data. I think that a paper that describes some publicly available data, either proposes new analyses or should be better sold as a review. In this paper the only analysis is a PBS (an Fst based statistics) on three modern human populations, chosen apparently arbitrarily. What is that supposed to test? How does that inform us on selection in modern humans after the split with archaic hominins? Assuming the test had power to test recent adaptations since the split of CEU and YRI, this would only capture a minimal fraction of the evolutionary time that occurred after humans separated from archaic humans. And the title of the paper is about Neanderthals and Denisovans, how come the only analysis is about three modern human populations? Also why those three populations? I found this part and the entire section about adaptive introgression very poorly described and motivated. Is there no introgressed overlapping these genes, or no introgressed haplotype previously shown to have signs of adaptive introgression overlapping those? In the first case, why not do tests to see if this is compatible with recent sweeps removing Neanderthal ancestry? All in all, there is only one statistical analysis in the paper, but this is not very informative. More should be done for a research paper in IJMS. For instance, how do the SIFT of fixed mutations for these genes compare with other comparable sets of genes? Can one get some estimates of the extent of purifying and positive selection, if any? I only saw percentages of missense mutations and so on but no tests on those, even basic ones like a MacDonal-Kreitman test, or basic comparisons with other sets of genes. There are also plenty of phylogenomic tools available (PAML etc.) that could be applied. I think the fundamental idea of the paper is nice, and clearly the authors have an in-depth understanding of the functions of these genes. However, the catalog should be the starting point, not the end one. Thus, I highly encourage the authors to incorporate some evolutionary analyses (not even necessarily the ones I suggested above) to raise the weight and improve the scientific completeness of the paper.   Minor comments: Fig.S2 is not very legible. I would make points bigger and darker.  

Author Response

AUTHORS' REPLIES ARE IN BOLD

A first point is that while the different genes are discussed lengthily in the paper, there does not seem to be an objective criterion for their inclusion. Are they all the differentially expressed genes in saliva, are they based on some GO? It feels arbitrary, despite the long introduction.

The lack of objective criteria also makes comparisons harder, particularly for statistical tests; though here this is paradoxically not a problem: there are little to no statistical tests. And here my major issue comes into play. The paper is pretty much devoid of evolutionary or genetic analyses. It is just a catalog out of publicly available data. I think that a paper that describes some publicly available data, either proposes new analyses or should be better sold as a review. In this paper the only analysis is a PBS (an Fst based statistics) on three modern human populations, chosen apparently arbitrarily. What is that supposed to test? How does that inform us on selection in modern humans after the split with archaic hominins? Assuming the test had power to test recent adaptations since the split of CEU and YRI, this would only capture a minimal fraction of the evolutionary time that occurred after humans separated from archaic humans. And the title of the paper is about Neanderthals and Denisovans, how come the only analysis is about three modern human populations? Also why those three populations? I found this part and the entire section about adaptive introgression very poorly described and motivated. Is there no introgressed overlapping these genes, or no introgressed haplotype previously shown to have signs of adaptive introgression overlapping those? In the first case, why not do tests to see if this is compatible with recent sweeps removing Neanderthal ancestry? All in all, there is only one statistical analysis in the paper, but this is not very informative. More should be done for a research paper in IJMS. For instance, how do the SIFT of fixed mutations for these genes compare with other comparable sets of genes? Can one get some estimates of the extent of purifying and positive selection, if any? I only saw percentages of missense mutations and so on but no tests on those, even basic ones like a MacDonal-Kreitman test, or basic comparisons with other sets of genes. There are also plenty of phylogenomic tools available (PAML etc.) that could be applied. I think the fundamental idea of the paper is nice, and clearly the authors have an in-depth understanding of the functions of these genes. However, the catalog should be the starting point, not the end one. Thus, I highly encourage the authors to incorporate some evolutionary analyses (not even necessarily the ones I suggested above) to raise the weight and improve the scientific completeness of the paper. Minor comments: Fig.S2 is not very legible. I would make points bigger and darker.  

We thank the Reviewer for his/her constructive feedbacks to our work. All the comments raised were duly taken into account in our revision and helped improving the overall quality of the manuscript.

We apologize for the lack of clarity here; we tried to improve the introduction and discussion sections explaining the reason why we have focused on these protein families. The analyzed salivary proteins represent the most abundant and best characterized protein species in saliva, and have been also implicated as biomarkers for clinical conditions, suggesting their significant contribution to oral homeostasis and salivary functions (please refer to lines 61-65 and 114-117 in the Introduction section and to lines 435-437 in the Discussion section of the revised manuscript).

Then, we explained more about why we included in our analysis Neanderthals from different areas, and which were the selection criteria for PBS analysis. Please refer to lines 577-582.

We thank the Referee for the insightful observations regarding the evolutionary analyses. To address these concerns, we chose to implement Tajima test as an additional evolutionary analysis to evaluate the selective effects of each observed substation. Accordingly, we have introduced this in the Results section (lines 403-407). “Tajima’s D values show comparable variance among the genes analyzed. The D values were prevalently slightly negative or positive (range from -0.698 to 3.359), confirming the absence of selective sweep, already suggested by PBS test.” File S3 reports the Tajima’s D values.

We also thank the Reviewer for the suggestions regarding Fig.S2 that have been modified accordingly.

Round 2

Reviewer 3 Report (New Reviewer)

I thank the authors for having tried to address my previous criticisms, by adding Tajima's D and amending a figure. I like the new title. Perhaps I would suggest "A catalog of", rather than "Identification": after all the selection of which genes to include is not based on some analysis, but on personal decision - even though biologically motivated.

Regarding PBS and the rationale behind the evolutionary analyses, I think they are partially addressed (also thanks to the change in title) but still leave me partially unsatisfied: CEU might be among the first populations to be included in the 1000 genome phase I, but long time passed since and if the goal of the paper is a comprehensive characterization of genetic variation in humans, more could have been done in this respect - even just including a more comprehensive representation of human genetic variation.

All in all I welcome the changes in the paper, and I think it could be considered for publication; though I confirm my assessment that the idea of the paper is nice, but the manuscript could be quite improved in terms of depth and quality of the analyses.

Author Response

The Authors sincerely appreciate the thoughtful feedbacks provided by the Reviewer.

In particular, we wish to thank him/her for the suggestion on how to improve the title, which  have been modified, accordingly.

Also, we may need to point out that we have used data from 1000 genomes to enable the alignment of the results with previous researches, and chose to use the European Ancestry sample since we treated Neanderthal's data. Bearing this issue in mind,  we performed the Tajama test with all the populations and using the unfiltered data to consider all the possible variability available to date.

This manuscript is a resubmission of an earlier submission. The following is a list of the peer review reports and author responses from that submission.

Round 1

Reviewer 1 Report

The presented manuscript gives an interesting overview of changes in the nucleotide sequence, namely nucleotide substitutions, that occurred in salivary proteins throughout the evolution. The authors have made a comprehensive analysis in order to annotate nucleotide variants of 17 salivary genes in the comparison between modern Homo sapiens and ancient hominis (Altai Neanderthal and Denisova). The Introduction provides good. The results are clearly presented and very well discussed. I compliment the authors on the work done and presented. The only thing I would recommend is adding a short separate conclusion presenting the main findings combined with the last paragraph of the Discussion. Since there are lot of findings mentioned in results and the discussion is thorough as well, such conclusion would improve the conveyed message.

Author Response

We thank the Reviewer for appreciating our work and for the suggestion regarding the conclusion. To this regard, in the revised version of the manuscript, we have introduced a separate conclusion section after the discussion one, and tried to better highlight the concluding remarks of the work.

Reviewer 2 Report

The study conducted by Di Pietro et al describe the divergence between H sapiens and archaic populations at genes expressed in saliva. The topic is certainly interesting, as it has been previously shown that H sapiens show strong divergences at these proteins when compared with other primates. The authors identify a large number of genetic variants that differ between humans and archaic populations, which in my opinion is important.

That said, I think the study is quite incomplete and it looks to me more like a report of changes between anatomically modern humans and archaic populations.

First of all, there is strong evidence of archaic introgression out of Africa (and even there). Some of the reported results suggest that at least some of the considered proteins show evidence of such archaic introgression (just by checking the allele frequencies). Authors could have checked the maps of archaic introgression already published (for example, the one from Akey, or the one conducted in Iceland) and check if there are signatures of archaic introgression for the considered genes. That would really give a meaning (or at least another more interesting meaning) to the title of their manuscript.

Second, authors could have checked the databases of recent positive selection in humans (popScan, for example) to see if some of the considered genes show evidence of positive selection in anatomically modern humans.

Finally, in my opinion the gene by gene description that the authors conduct provides very little information of the overall evolutionary process in these genes (namely, archaic introgression, positive selection, purifying selection and neutrality). It would be more interesting if the authors summarize all the results towards explaining an evolutionary story of the salivary proteins without going into the details of particular genes.

The last reasoning also extends to the introduction, where different genes are profusely explained. I do not think there is such a need, if the scope is understanding the evolution of salivary proteins as a whole. The different roles could just be summarized in a single paragraph without reducing the amount of required information for understanding the content of the study.

Other points: There are three Neanderthal sequences that the authors could use: Altai 8 (used in the manuscript), Vindija and Chagyrskya. It is known that these three individuals show genetic differences suggesting population substructure within the Neanderthals. I think it would be desirable to include them in the study, given the title of the manuscript.

Author Response

All the co-Authors wish to thank the reviewer for his/her precious suggestions and comments, which we have tried to address including additional experimental activities, thanks to the collaboration with experts in the field, who were added in the Authors' list. We believe that the manuscript has significantly improved its quality and overall message thanks to this implementation and revision.
